# An *Arabidopsis* AT-hook motif nuclear protein mediates somatic embryogenesis and coinciding genome duplication

Omid Karami[1✉], Arezoo Rahimi[1], Patrick Mak[1,4], Anneke Horstman [2,3], Kim Boutilier[2], Monique Compier[1,5], Bert van der Zaal[1] & Remko Offringa [1✉]

Plant somatic cells can be reprogrammed into totipotent embryonic cells that are able to form differentiated embryos in a process called somatic embryogenesis (SE), by hormone treatment or through overexpression of certain transcription factor genes, such as *BABY BOOM* (*BBM*). Here we show that overexpression of the *AT-HOOK MOTIF CONTAINING NUCLEAR LOCALIZED 15* (*AHL15*) gene induces formation of somatic embryos on *Arabidopsis thaliana* seedlings in the absence of hormone treatment. During zygotic embryogenesis, *AHL15* expression starts early in embryo development, and *AH15* and other *AHL* genes are required for proper embryo patterning and development beyond the globular stage. Moreover, *AHL15* and several of its homologs are upregulated and required for SE induction upon hormone treatment, and they are required for efficient *BBM*-induced SE as downstream targets of BBM. A significant number of plants derived from *AHL15* overexpression-induced somatic embryos are polyploid. Polyploidisation occurs by endomitosis specifically during the initiation of SE, and is caused by strong heterochromatin decondensation induced by *AHL15* overexpression.

[1] Plant Developmental Genetics, Institute of Biology Leiden, Leiden University, Leiden, Netherlands. [2] Bioscience, Wageningen University and Research, Wageningen, Netherlands. [3] Laboratory of Molecular Biology, Wageningen University and Research, Wageningen, Netherlands. [4] Present address: Sanquin Plasma Products B.V., Amsterdam, Netherlands. [5] Present address: Rijk Zwaan Netherlands B.V., De Lier, The Netherlands. ✉email: o.karami@biology.leidenuniv.nl; r.offringa@biology.leidenuniv.nl

The conversion of somatic cells into embryonic stem cells is a process that occurs in nature in only a few plant species, for example on the leaf margins of *Bryophyllum calycinum*[1] or *Malaxis paludosa*[2], or from the unfertilized egg cell or ovule cells of apomictic plants[3,4]. By contrast, for many more plant species, somatic cells can be converted into embryonic cells under specific laboratory conditions[5,6]. The process of inducing embryonic cell fate in somatic plant tissues is referred to as somatic embryogenesis (SE). Apart from being a tool to study and understand early embryo development, SE is also an important tool in plant biotechnology, where it is used for asexual propagation of (hybrid) crops or for the regeneration of genetically modified plants during transformation[7].

SE is usually induced in in vitro cultured tissues by exogenous application of plant growth regulators. A synthetic analog of the plant hormone auxin, 2,4-dichlorophenoxyacetic acid (2,4-D), is the most commonly used plant growth regulator for the induction of SE[8,9]. During the past two decades, several genes have been identified that can induce SE on cultured immature zygotic embryos or seedlings when overexpressed in the model plant *Arabidopsis thaliana*[6,10]. Several of these genes, including *BABY BOOM* (*BBM*) and *LEAFY COTYLEDON 1* (*LEC1*) and *LEC2*, have now been recognized as key regulators of SE[11–14].

Here we show that overexpression of *Arabidopsis AT-HOOK MOTIF CONTAINING NUCLEAR LOCALIZED 15* (*AHL15*) can also induce somatic embryos (SEs) on germinating seedlings in the absence of plant growth regulators. AT-hook motifs exist in a wide range of eukaryotic nuclear proteins, and are known to bind to the narrow minor groove of DNA at short AT-rich stretches[15,16]. In mammals, AT-hook motif proteins are chromatin modification proteins that participate in a wide array of cellular processes, including DNA replication and repair, and gene transcription leading to cell-growth, cell-differentiation, cell-transformation, cell-proliferation, and cell-death[17]. The *Arabidopsis* genome encodes 29 AHL proteins that contain one or two AT-hook motifs and a plants and prokaryotes conserved (PPC) domain, that directs nuclear localization and contributes to the physical interaction of AHL proteins with other nuclear proteins, such as transcription factors[18,19]. *AHL* gene families are found in angiosperms and also in early diverging land plants such as *Physcomitrella patens* and *Selaginella moellendorffii*[20,21]. *Arabidopsis* AHL proteins have roles in several aspects of plant growth and development, including flowering time, hypocotyl growth[20,21], flower development[22], vascular tissue differentiation[23], and gibberellin biosynthesis[24]. Recently, we have shown that *AHL15* also enhances plant longevity by suppressing axillary meristem maturation[25]. How plant AHL proteins regulate these underlying biological events is largely unknown. Here, we show that *AHL15* and its homologs play major roles in directing plant cell totipotency during both zygotic embryogenesis and 2,4-D-mediated and BBM-mediated SE. Furthermore, our data show that AHL15 has a role in chromatin opening, and that its overexpression induces SE coinciding with endomitosis and polyploidy.

## Results

**Overexpression of *AHL* genes induces SE**. *AHL15* was originally identified in a yeast one-hybrid screen, and functional analysis revealed that overexpression of the gene in *Arabidopsis thaliana* (Arabidopsis) under control of the *35S* promoter (*p35S:AHL15*) results in induction of SE on germinating seedlings[26]. *AHL15* overexpression seedlings initially remained small and pale and then developed very slowly (Fig. 1a). Three to four weeks after germination, seedlings from the majority of the transgenic lines (41 of 50 lines) recovered from this growth retardation (Fig. 1a)

and underwent relatively normal development, producing rosettes, flowers, and finally seeds. However, in the remaining *p35S:AHL15* lines (9 of 50 lines), globular structures could be observed on seedling cotyledons 1–2 weeks after germination (Fig. 1b). These structures developed into heart-shaped or torpedo-shaped somatic embryos (Fig. 1c) that could be germinated to produce fertile plants. SE was not observed in other tissues, such as roots, hypocotyls or leaves (Fig. 1b).

In Arabidopsis, the cotyledons of immature zygotic embryos (IZEs) are the most competent tissues for SE in response to the synthetic auxin 2,4-dichlorophenoxyacetic acid (2,4-D)[8]. Remarkably, a high percentage (85–95%) of the IZEs from the nine selected *p35S:AHL15* lines were able to produce somatic embryos when cultured on medium lacking 2,4-D. When left for a longer time on this medium, these primary *p35S:AHL15* somatic embryos produced secondary somatic embryos (Fig. 1d, e), and in two of the nine *p35S:AHL15* lines, this repetitive induction of SE resulted in the formation of embryonic masses (Fig. 1f). Overexpression of other *Arabidopsis AHL* genes encoding proteins with a single AT-hook motif (i.e., the closest paralogs *AHL19* and *AHL20*, and *AHL29* as a more distant one; Supplementary Fig. 1), did not induce SE on germinating seedlings, but did induce SE on a low percentage (20–30%) of the IZEs in the absence of 2,4-D (Supplementary Fig. 2). These results suggest that AHL proteins can enhance the embryonic competence of plant tissues, with AHL15 being able to most efficiently induce a totipotent state without addition of 2,4-D.

**AHL genes are important during zygotic embryogenesis**. Given the role of *AHL* genes in promoting in vitro totipotency, we determined whether these genes also have a role during zygotic embryogenesis. Expression analysis using the *pAHL15:AHL15-GUS* and *pAHL15:AHL15-tagRFP* lines, both in the wild-type background, showed that *AHL15* is expressed in zygotic embryos (ZEs) from the four cell embryo stage onward (Fig. 2a–i). In line with the previously reported nuclear localization of AHL proteins[22,27], the AHL15-tagRFP fusion protein was detected in the nucleus (Fig. 2e–i and Supplementary Fig. 3). Single *ahl15* or *ahl19* loss-of-function mutants or *ahl15 ahl19* double mutants carrying an artificial microRNA targeting *AHL20* (*ahl15 ahl19 amiRAHL20*) showed wild-type ZE development (Supplementary Fig. 4). Also *pAHL15:AHL15-GUS* plants produced wild-type embryos (Fig. 2n) and seeds (Fig. 2j). However, in reciprocal crosses of this reporter line with the *ahl15* mutant we were unable to obtain homozygous *ahl15 pAHL15:AHL15-GUS* seedlings among 50 F2 plants that were genotyped. Irrespective of the direction in which the cross was made, F1 siliques showed a wild-type phenotype, whereas siliques of *ahl15/+ pAHL15:AHL15-GUS* F2 plants contained around 25% brown, shrunken seeds (Fig. 2k and Supplementary Fig. 5) that were unable to germinate. Embryos in these shrunken seeds showed patterning defects and did not develop past the globular stage (Fig. 2o). These results suggested that the AHL15-GUS fusion protein has a strong dominant negative effect on AHL function in the absence of the wild-type AHL15 protein. This seemed specific for the AHL15-GUS fusion, as fertile homozygous *ahl15 pAHL15:AHL15-tagRFP* plants showing wild-type development could be obtained for three independent *pAHL15:AHL15-tagRFP* lines.

To confirm that the chimeric AHL15-GUS fusion protein caused the mutant phenotypes only in the mutant background, we introduced the *pAHL15:AHL15-GUS* construct in the *ahl15 pAHL15:AHL15* background. ZEs of *ahl15 pAHL15:AHL15* plants did not show any morphological defects (Fig. 2l), and the resulting *ahl15 pAHL15:AHL15-GUS pAHL15:AHL15* siliques showed normal seed development (Fig. 2m). These results

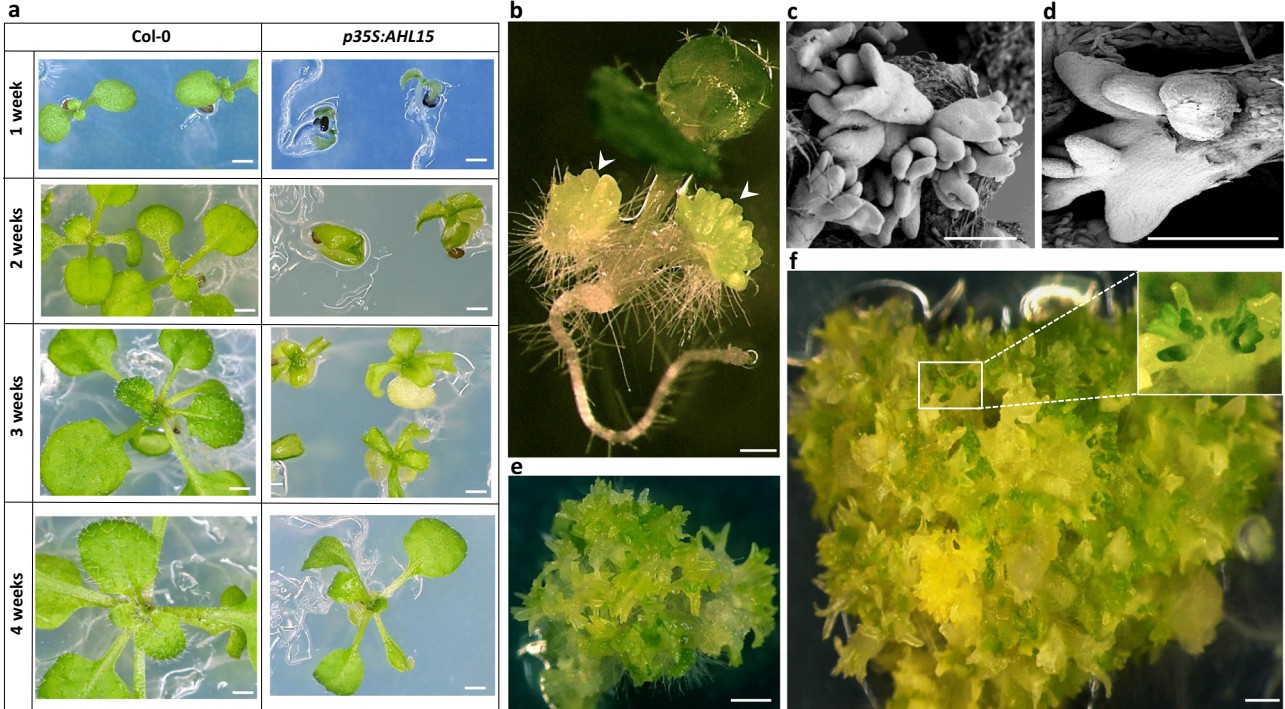

**Fig. 1 Overexpression of _AHL 15_ delays _Arabidopsis_ seedling development and induces SE. a** The morphology of 1-week-old, 2-week-old, 3-week-old, and 4-week-old wild-type and _p35S:AHL15_ seedlings and plants grown in long day conditions (16 h photoperiod). **b** Two-week-old _p35S:AHL15 Arabidopsis_ seedling with somatic embryos on the cotyledons (arrowheads). **c, d** Scanning electron micrographs showing torpedo stage somatic embryos on _p35S: AHL15_ cotyledons (**c**) or the secondary somatic embryos formed on a _p35S:AHL15_ primary somatic embryo (**d**). **e** The morphology of a 3-week-old _AHL15_ overexpression-induced embryonic mass following secondary SE. **f** The morphology of a 2-month-old embryonic mass formed from a _p35S:AHL15_ seedling. Size bars indicate 0.5 cm (**a**) and 1 mm (**b-f**). **a-f** Similar results were obtained from four independent experiments.

indicate that wild-type AHL15, when sufficiently expressed, is able to negate the dominant negative effect of the AHL15-GUS fusion.

AHL proteins bind to each other through their PPC domain and form complexes with non-AHL transcription factors through a conserved six-amino-acid region in the PPC domain. In Arabidopsis, expression of an AHL protein without the conserved six-amino-acid region in the PPC domain leads to a dominant negative effect, as the truncated protein can still bind to other AHL proteins, but creates a non-functional complex[19]. Based on this finding and a similar dominant-negative approach reported for human AT-Hook proteins[28], we deleted these six amino acids from the PPC domain of AHL15 (Fig. 2p) and expressed it under the _AHL15_ promoter (_pAHL15:AHL15-ΔG_) in wild-type plants. All plant lines generated with a _pAHL15:AHL15-ΔG_ construct ($n = 20$) were fertile and developed wild-type ZEs (Fig. 2q). However, like for the _ahl15/+ pAHL15:AHL15-_GUS lines, also _ahl15/+ pAHL15:AHL15-ΔG_ plants showed defective seed development (Fig. 2r) and were unable to produce homozygous _ahl15 pAHL15:AHL15-ΔG_ progeny, indicating that this genetic combination is also embryo lethal. These results provide additional support for the dominant negative effect caused by the AHL15-GUS fusion protein, showing that AHL15 and homologs are important during zygotic embryogenesis.

**_AHL_ genes are required for 2,4-D-induced and BBM-induced SE.** Next we investigated the contribution of _AHL_ genes to 2,4-D-induced SE, by culturing IZEs from _ahl_ loss-of-function mutants on medium containing 2,4-D. Only a slight reduction in SE induction efficiency was observed in the single _ahl15_ loss-of-function mutant (Fig. 3a), which triggered us to examine the contribution of other _AHL_ genes in this process. RT-qPCR

analysis showed that _AHL15, AHL19_, and _AHL20_ expression was significantly upregulated in IZEs following 7 days of 2,4-D treatment (Fig. 3b). Moreover, analysis of _pAHL15:AHL15-GUS_ expression in IZEs showed that _AHL15_ expression was specifically enhanced in the cotyledon regions where somatic embryos are initiated (Fig. 3c, d). IZEs from triple _ahl15 ahl19 amiRAHL20_ mutants produced significantly less somatic embryos (Fig. 3e). Moreover, whereas _pAHL15:AHL15-GUS_ IZEs showed a near wild-type capacity to form somatic embryos in the presence of 2,4-D, a strong decrease in the embryogenic capacity of the IZE explants was observed when the _pAHL15:AHL15-GUS_ reporter was in the _ahl15/+_ mutant background (Fig. 3a). The majority of the IZEs produced non-embryogenic calli that were not observed in the other genotypes (Fig. 3e). These results indicate that _AHL15_ and several homologs are required for 2,4-D-induced somatic embryo formation starting from IZEs.

Overexpression of the AINTEGUMENTA-LIKE (AIL) transcription factor BBM efficiently induces SE in in the absence of exogenous growth regulators[13,29]. Genome-wide analysis of BBM binding sites using chromatin immunoprecipitation (ChIP)[30] in 2,4-d and _p35S:BBM_-induced somatic embryos showed significant BBM binding to the proximal promoter (~ −200 to −500 bp) of _AHL15, AHL19_, and _AHL20_ (Fig. 3f–i), suggesting that these _AHL_ genes are direct downstream BBM targets. Analysis of gene expression changes in _p35S:BBM-GR_ plants 3 h after treatment with dexamethasone (DEX) and the translational inhibitor cycloheximide (CHX) showed that BBM activated the expression of _AHL15_ and _AHL20_. A slight but statistically not significant enhancement in expression ($p = 0.1$) was observed for _AHL19_ (Fig. 3j). As expected, no difference in expression ($p = 0.9$) was observed for _AHL29_ (Fig. 3j), which proximal promoter was not bound by BBM (Fig. 3i).

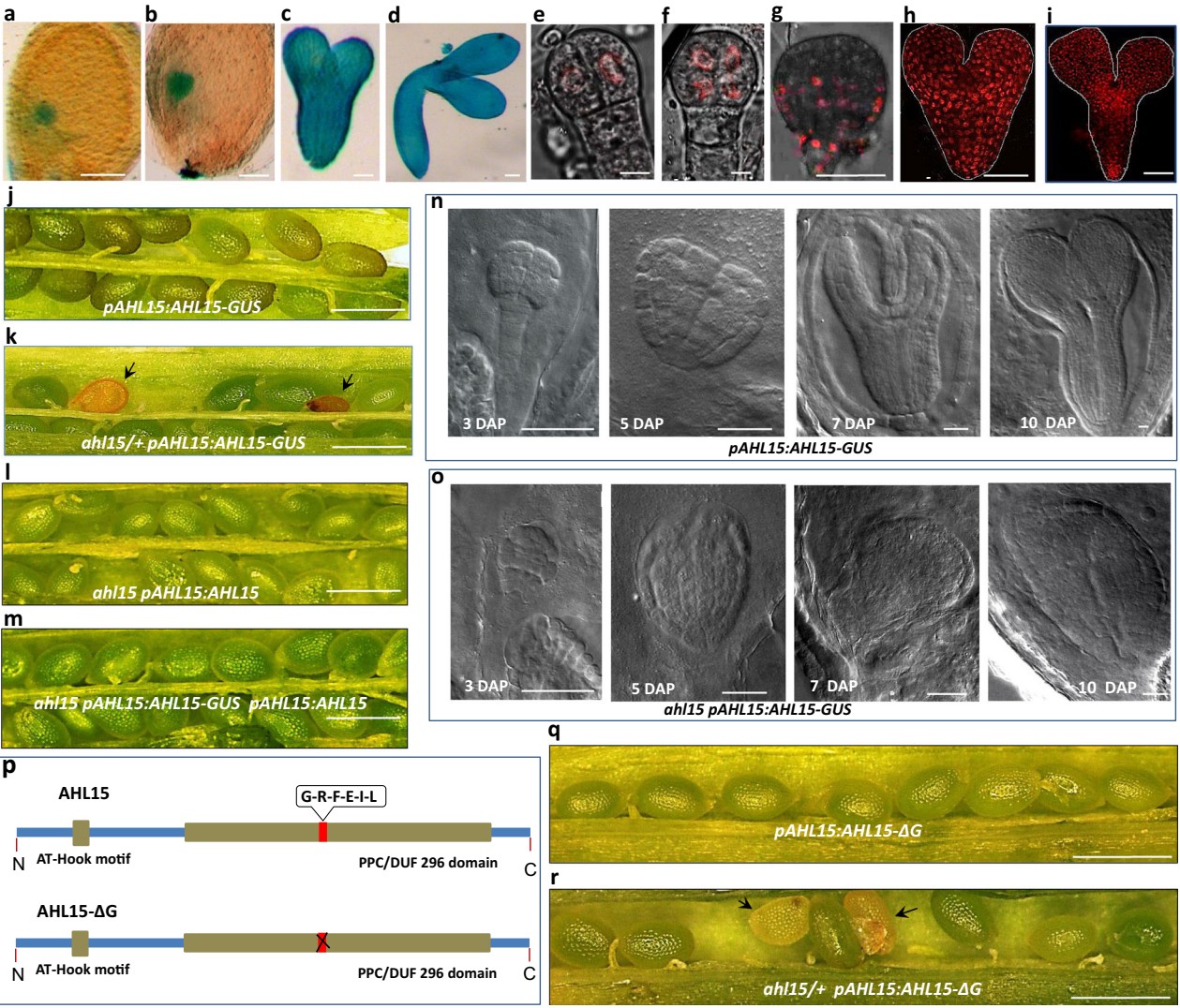

**Fig. 2 *AHL15* is expressed and essential during ZE. a–d** Expression pattern of *pAHL15:AHL15-GUS* in globular-stage (**a**), heart-stage (**b**), torpedo-stage (**c**) and bent cotyledon (**d**) stage embryos. **e–i** Confocal microscopy images of *pAHL15:AHL15-tagRFP* four cell (**e**), 8 cell-stage (**f**), globular-stage (**g**), heart-stage (**h**), and torpedo-stage (**i**) embryos. **j** Wild-type seed development in a *pAHL15:AHL15-GUS* silique. **k** Aberrant seed development (arrowheads) in a *ahl15/+ pAHL15:AHL15-GUS* silique. **l, m** Wild-type seed development in *ahl15 pAHL15:AHL15* (**l**) or *ahl15 pAHL15:AHL15 pAHL15:AHL15-GUS* (**m**) siliques. **n, o** DIC images of zygotic embryo development in siliques of *pAHL15:AHL15-GUS* (**n**) or *ahl15/+ pAHL15:AHL15-GUS* (**o**) plants at 3, 5, 7, or 10 days after pollination (DAP). **p** Schematic domain structure of AHL15 and the dominant negative AHL15-ΔG version, in which six-conserved amino-acids (Gly-Arg-Phe-Glu-Ile-Leu, red box) are deleted from the C-terminal PPC domain. **q** Wild-type seed development in *pAHL15:AHL15-ΔG* siliques. **r** Aberrant seed development (arrowheads) in ahl15/+ *pAHL15:AHL15-ΔG* siliques (observed in three independent *pAHL15:AHL15-ΔG* lines crossed with the *ahl15* mutant). **a–r** Similar results were obtained from two independent experiments. Size bar indicates 100 µm (**a–d**, **g–l**), 10 µm (**e**, **f**), 1 mm (**j–m, q, r**) and 40 µm (**n, o**).

Next, we investigated the requirement for *AHL* genes in BBM-induced SE by comparing the effectiveness of the *p35S:BBM-GR* construct in inducing SE in the wild-type or *ahl15 ahl19 amiRAHL20* triple mutant background. DEX treatment induced SE in 40 of the 554 primary *p35S:BBM-GR* transformants (7%) in the wild-type background, but this was completely abolished in the *ahl15 ahl19 amiRAHL20* background (0 of the 351 primary transformants). These results, together with the observation that *AHL15* overexpression, like *BBM* overexpression induces spontaneous SE, suggest that induction of *AHL* gene expression is a key regulatory component of the BBM signaling pathway.

**AHL15 overexpression-mediated chromatin decondensation correlates with SE induction.** Based on the observation in animal cells that AT-hook proteins are essential for the open chromatin in neural precursor cells[31–33], we investigated whether AHL15

modulates the chromatin structure during SE initiation. As a measure for global chromatin structure we quantified the amount of tightly condensed, transcriptionally-repressed regions (heterochromatin), which can be visualized using fluorescent chromatin markers or DNA staining. In somatic plant cells, large scale changes in heterochromatin have been associated with cell identity reprogramming[34,35].

First, by tracking SE induction on *p35S:AHL15* IZEs, we observed that protodermal cells at the adaxial side of cotyledons started to divide around 6 days after culture (Supplementary Fig. 6A), leading to the formation of *pWOX2:NLS-GFP* expressing pro-somatic embryos (Supplementary Fig. 6B). Propidium iodide (PI) staining of chromosomal DNA in cotyledon protodermal cells of *35S:AHL15* IZEs showed a remarkable dispersion of heterochromatin coinciding with the appearance of *pWOX2:NLS-GFP* expressing pro-somatic embryos at 7 days after culture

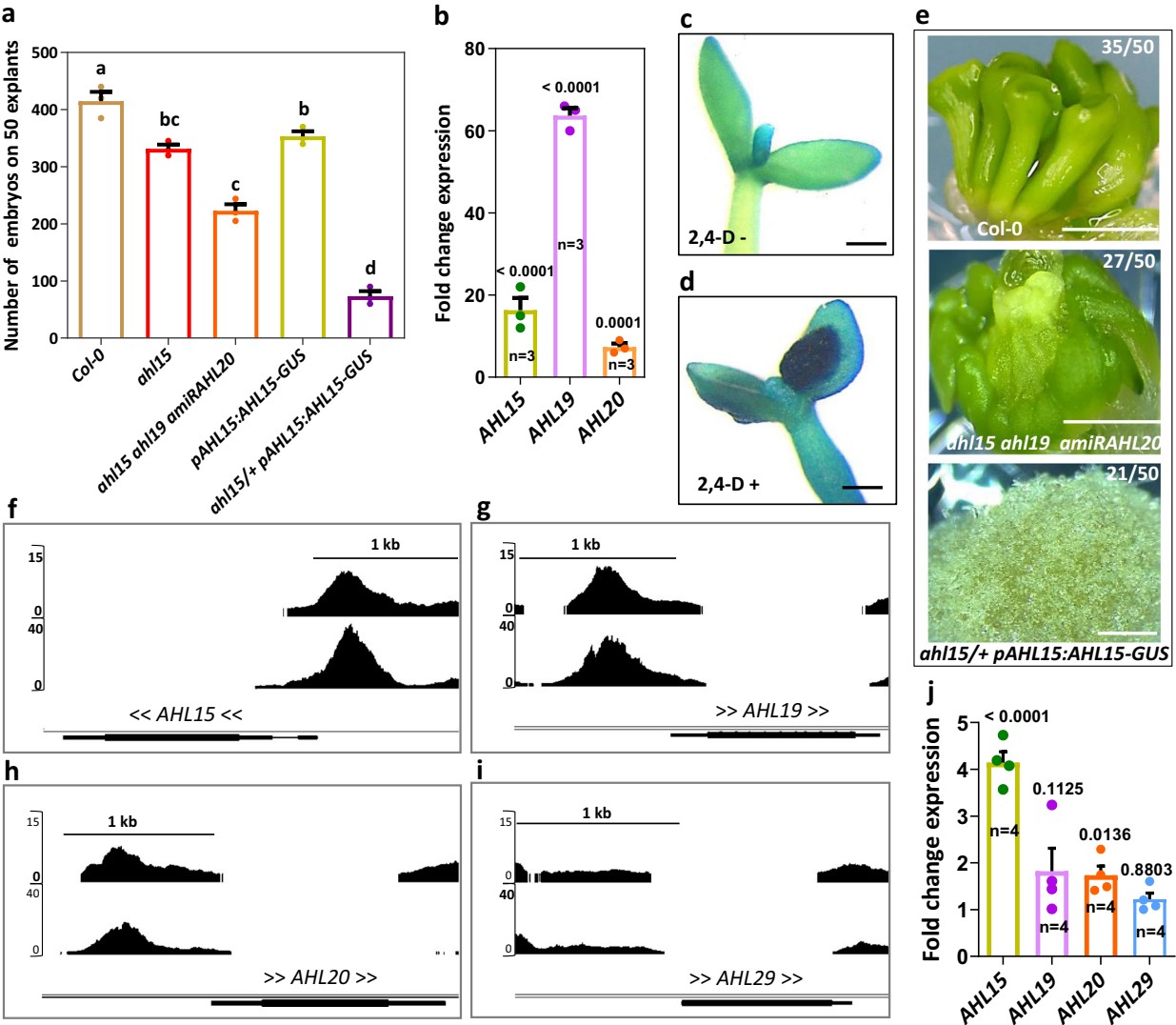

**Fig. 3 *AHL* genes are essential for 2,4-D-induced and BBM-induced SE. a** The effect of *ahl* loss-of-function on the capacity to induce somatic embryos on IZEs by 2,4-D. Dots indicate the values of three biological replicates per plant line with 50 IZEs per replicate, bar indicates the mean, and error bars the s.e. m. For the *ahl15/+ pAHL15:AHL15-GUS* combination IZEs were harvested from *ahl15/+ pAHL15:AHL15-GUS* plants and scored and genotyped after culture. Only scores of *ahl15/+ pAHL15:AHL15-GUS* explants were used. The letters indicate statistically different values, as determined by a one-way ANOVA with Tukey's post hoc test. The *p*-values are provided in the source data file. **b** RT-qPCR analysis of the fold change in expression of *AHL15*, *AHL19* and *AHL20* in IZEs cultured for 7 days on medium with 5 μM 2,4-D relative to medium without 2,4-D. **c, d** *pAHL15:AHL15-GUS* IZEs cultured for 7 days in the absence (**c**) or presence (**d**) of 5 μM 2,4-D and histochemically stained for GUS activity. Similar results were obtained from two independent experiments. Size bar indicates 1 mm. **e** Predominant phenotype of wild-type (upper panel), *ahl15 ahl19 amiRAHL20* (middle panel), or *ahl15/+ pAHL15:AHL15-GUS* (lower panel) IZEs cultured for 2 weeks on 2,4-D medium. Numbers indicate the frequency with which similar phenotypes were observed in two independent experiments. Size bar indicates 1 mm. **f–i** ChIP-seq BBM binding profiles for *AHL15* (**f**), *AHL19* (**g**), *AHL20* (**h**), and *AHL29* (**i**). The binding profiles from the *pBBM:BBM-YFP* (upper profile) and *p35S:BBM-GFP* (lower profile) ChIP-seq experiments are shown. The *x*-axis shows the position of DNA binding relative to the location of the selected gene (TAIR 10 annotation), the *y*-axis shows the ChIP-seq score (fold enrichment of the BBM-GFP/YFP ChIP to the control ChIP), and the arrow brackets around the gene name indicate the direction of gene transcription. **j** RT-qPCR analysis of the fold change in expression of *AHL* genes in DEX + CHX treated *p35S:BBM-GR* seedlings relative to that in DEX + CHX treated Col-0 wild-type seedlings. **b, j** Dots indicate the values of three or four biological replicates, bar indicates the mean, and error bars the s.e.m. The *p*-values were determined by a two-sided Student's *t*-test.

(Fig. 4a and Supplementary Fig. 7). In contrast, cotyledon protodermal cells of wild-type IZEs did not show a clear change in heterochromatin state at this time point (Fig. 4a and Supplementary Fig. 7). Based on the fraction of compacted chromatin per nucleus, we categorized nuclei into either the condensed (Supplementary Fig. 8a) or the dispersed phenotype (Supplementary Fig. 8b). Quantification showed that the percentage of dispersed nuclei highly increased in cotyledon

protodermal cells of *35S:AHL15* IZEs after 7 days of culture relative to wild-type IZEs (Fig. 4b).

The *Arabidopsis* HISTONE 1.1-GFP[36] and HISTONE 2B-GFP[35] proteins are incorporated into nucleosomes, providing markers for the chromatin state in living cells. H1.1-GFP and H2B-GFP fluorescence observations confirmed that the chromo-centers in cotyledon protodermal cells of 7-day-cultured *35S: AHL15* IZEs (Fig. 4c, d) were much more diffuse compared to

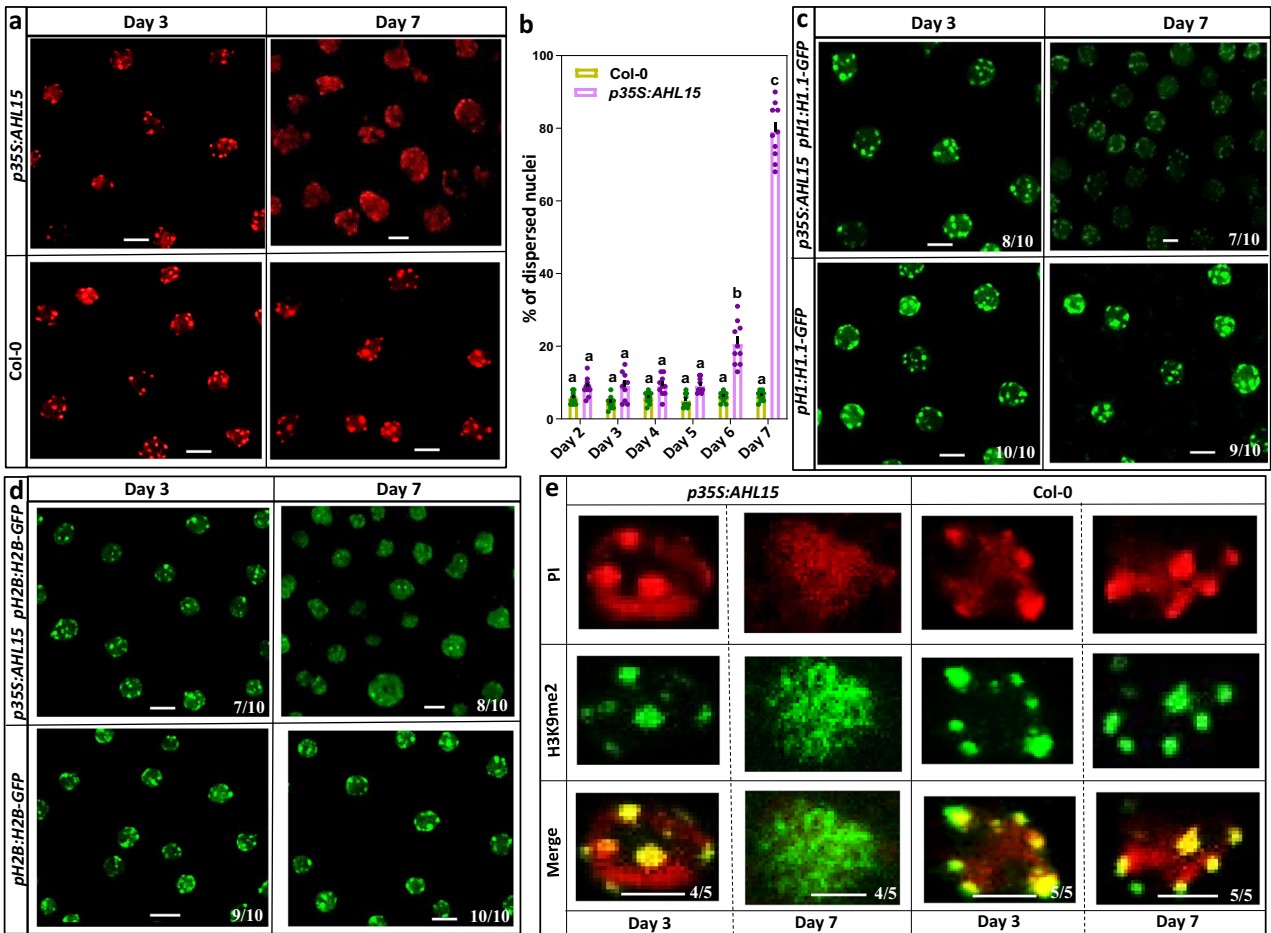

**Fig. 4 AHL15 overexpression reduces heterochromatin condensation during SE induction. a** Visualization of DNA compaction using propidium iodide (PI) staining in cotyledon protodermal nuclei of wild-type and *p35S:AHL15* IZEs 3 or 7 days after culture. **b** Quantification of the percentage of dispersed PI-labeled nuclei (**a**) according to the classification shown in Supplementary Fig. 7. Dots indicate the values of ten biological replicates per plant line with about 200 nuclei analyzed per replicate, bar indicates the mean, and error bars the s.e.m. Different letters indicate statistically significant differences as determined by one-way ANOVA with Tukey's honest significant difference post-hoc test. The *p*-values are provided in the source data file. **c**, **d** Visualization of DNA compaction in H1.1-GFP (**c**) or H2B-GFP (**d**) labeled nuclei in cotyledon protodermal cells of wild-type and *p35S:AHL15* IZEs 3 or 7 days after culture. **e** Visualization of heterochromatin compaction in nuclei of cotyledon protodermal cells stained with PI or immunostained against the heterochromatin marker H3K9me2 in wild-type and *p35S:AHL15* IZEs 3 or 7 days after culture. **a**, **c–e** Size bar indicates 6 μm and numbers in images indicate the frequency with which similar images from different samples were obtained in two independent experiments.

cells of 3-day-cultured IZEs (Fig. 4c, d). No significant differences in H1.1-GFP and H2B-GFP signals were detected between cotyledon protodermal cells of three and seven day-incubated wild-type IZEs (Fig. 4c, d).

To further confirm the changes in the heterochromatin state in cotyledon cells, we analyzed the dynamic localization of the heterochromatin mark H3K9me2 by immunostaining[37,38]. Dispersal of the PI stain in nuclei of cotyledon protodermal cells of *35S:AHL15* IZEs 7 days after culture correlated very well with the dispersed H3K9me2 signal (Fig. 4e and Supplementary Fig. 9). In contrast, both the PI staining and the H3K9me2 signal remained condensed in wild-type nuclei. Together these data indicate that AHL15 promotes heterochromatin decondensation. Surprisingly, in cells expressing both *pAHL15:AHL15-tagRFP* and *pH2B:H2B-GFP* reporters, *AHL15-tagRFP* did not co-localize with the chromocenters (Supplementary Fig. 10), but showed a more diffuse nuclear distribution, suggesting that AHL15 action is not limited to heterochromatin, but that the protein rather regulates global chromatin decondensation.

To explore whether the observed heterochromatin decondensation in *35S:AHL15* IZEs also occurs in other SE systems, we

investigated the heterochromatin state in embryonic cells induced by 2,4-D. The H1.1-GFP signals in cotyledon protodermal cells of *pH1.1:H1.1-GFP* IZEs cultured for 7 days on medium containing 2,4-D displayed only moderate decondensation of heterochromatin, with chromocenters becoming significantly smaller in cotyledon cells after 2,4-D treatment (Supplementary Fig. 11a–c), but not showing the diffuse signal observed in *p35S:AHL15* IZEs (Fig. 4c). This suggests that the AHL15-induced chromatin decondensation is not a general trait of cells undergoing SE, but rather, is specific for AHL15-overexpressing cells. Although 2,4-D strongly upregulates *AHL15* expression during somatic embryogenesis (Fig. 3b–d), *35S* promoter-driven *AHL15* expression resulted in approximately 3-fold higher mRNA levels than observed with 2,4-D treatment (Supplementary Fig. 11d). This difference in expression levels might explain the strong chromatin decondensation observed in *35S::AHL15* IZEs compared to 2,4-D-treated IZEs.

To obtain insight into the role of AHL15 in chromatin decondensation during zygotic embryogenesis, we introduced the *pH2B:H2B-GFP* reporter into the *ahl15/+ pAHL15:AHL15-GUS* background. In defective *ahl15 pAHL15:AHL15-GUS* embryos,

**Table 1 Ploidy level of plants derived from SEs induced by _AHL15_ overexpression, _BBM_ overexpression or by 2,4-D treatment.**

| Genotype | SE-derived plants | Ploidy level of plants[a] | | | Ploidy percentage |
|---|---|---|---|---|---|
| | | 2n | 4n | 8n | |
| _p35S:AHL15-2_ | 16 | 5 | 11 | – | 69 |
| _p35S:AHL15-4_ | 6 | 4 | 1 | 1 | 33 |
| _p35S:AHL15-13_ | 11 | 7 | 4 | – | 36 |
| _p35S:AHL15-14_ | 17 | 14 | 2 | 1 | 18 |
| _p35S:AHL15-15_ | 15 | 11 | 4 | – | 27 |
| Wild type, 2-4-D | 30 | 30 | – | – | 0 |
| _p35S:BBM_ | 20 | 20 | – | – | 0 |

[a]The ploidy level was analyzed by counting the chloroplast number in guard cells. For plants derived from _p35S:AHL15_-induced SEs, the ploidy level was confirmed using flow cytometry (Supplementary Fig. 15).

we observed irregular shaped chromocenters that were much larger than those in wild-type cells (Supplementary Fig. 12). This result, together with the reduced heterochromatin condensation observed in cotyledon cells of _p35S:AHL15_ IZEs, suggests that _AHL15_ plays a role in regulating the chromatin architecture during embryogenesis.

**_AHL15_ overexpression induces polyploidy during SE initiation**. Plants regenerated from somatic embryos derived from _p35S: AHL15_ IZEs (without 2,4-D treatment) developed large rosettes with dark green (Supplementary Fig. 13) leaves and large flowers (Supplementary Fig. 14a). These phenotypes were not observed in _p35S:AHL15_ progeny obtained through zygotic embryogenesis. As these phenotypes are typical for polyploid plants, we determined the ploidy level of the plants. The number of chloroplasts in guard cells[39] of plants showing large flowers was two times higher (8–12) than that of diploid wild-type plants (4–6) (Supplementary Fig. 14b). Polyploidisation is also correlated with an increase in cell size and nuclear size in _Arabidopsis_ and many other organisms[40]. Indeed root cells of tetraploid _35S::AHL15_ seedlings showed a larger nucleus than diploid control plants (Supplementary Fig. 14c). Using the centromere-specific HISTONE3-GFP fusion protein (CENH3-GFP)[41,42], seven to eight CENH3-GFP-marked centromeres could be detected in root cells of wild-type Arabidopsis plants and diploid _35S:AHL15_ SE-derived plants (Supplementary Fig. 12d). By contrast, around 12–16 centromeres were observed in the larger nuclei in root cells of tetraploid _p35S:AHL15_ plants (Supplementary Fig. 14d). This confirmed that the plants with large organs that were regenerated from _AHL15_ overexpression-induced somatic embryos are polyploid. Additional flow cytometry analysis on SE-derived plant lines confirmed that most of these plants were tetraploid, and two were octoploid (Table 1 and Supplementary Fig. 15). The frequency of SE-derived polyploidy varied per _p35S:AHL15_ line, ranging from 18 to 69% (Table 1). This variety among lines most likely relates to the level of _AHL15_ overexpression in the different _p35S:AHL15_ lines. No polyploid plants were obtained from somatic embryos induced by 2,4-D on wild-type IZEs, or by _BBM_ overexpression in seedlings (Table 1), indicating that polyploidisation is specifically induced by _AHL15_ overexpression.

The considerable frequency of polyploid plants regenerated from _p35S:AHL15_ somatic embryos posed the question as to when polyploidisation is induced, and whether it is correlated with, or even promoted by SE induction. We observed a variable number of CENH3-GFP labeled centromeres (6–8, 12–15, and 25–30) in nuclei of protodermal cells at the adaxial side of of _p35S:AHL15_ IZE cotyledons 7–8 days after of the start of culture, coinciding with SE induction and reflecting the presence of diploid, tetraploid, and octoploid cells (Fig. 5a). Quantitative analysis showed that nuclei with more than 11 detectable

centromeres were only observed in cotyledon protodermal cells of _p35S:AHL15_ IZEs after 7 days of culture (Fig. 5b), demonstrating that polyploidisation coincides with SE induction. No evidence was obtained for polyploidy in root meristems (Supplementary Fig. 16a) or young leaves (Supplementary Fig. 16b) of ZE-derived _p35S:AHL15_ plants, nor was polyploidy observed in the 2,4-D-induced non-embryogenic calli found on leaf and root tissues of _p35S:AHL15_ plants (Supplementary Fig. 16c, d). When we followed the _pH2B:H2B-GFP_ reporter in cotyledons of _p35S:AHL15_ IZEs in time, we did not observe any cells with an increased number of chromocenters during the first week of culture (Fig. 5c). At 7 days of IZE culture, however, an increase in chromocenter number could be detected in proliferating _p35S:AHL15_ cotyledon cells (Fig. 5d). This result showed that polyploidisation in _p35S:AHL15_ cotyledon cells is tightly associated with the induction of SE. Based on these results, and in line with the observation that _p35S:AHL15_ polyploid plants were only obtained from _p35S:AHL15_ somatic embryos, we conclude that the _AHL15_-induced polyploidisation occurs specifically during somatic embryo initiation.

**_AHL15_ overexpression-induced polyploidy occurs by endomitosis due to chromosome mis-segregation**. Endoreduplication normally occurs in expanding cells to facilitate cell growth. During endoreduplication, duplicated chromosomes do not enter into mitosis and the number of chromocenters does not increase[43–45]. We observed an increase in H2B-GFP-marked chromocenters in _p35S:AHL15_ cotyledon cells that coincided with polyploidisation events (Fig. 5c, d), but not in endoreduplicated nuclei of wild-type root hair, leaf, or root cells (Fig. 5e–g), suggesting that these polyploid _p35S:AHL15_ cells are not derived from endoreduplication. Thus, duplication of segregated chromosomes in _35S:AHL15_ cotyledons cells must be caused by endomitotis, during which mitosis is initiated and chromosomes are separated, but cytokinesis fails to occur. Although ectopic overexpression of _AHL15_ resulted in a high percentage of polyploid plants (Table 1), polyploidy of the embryo itself was not a prerequisite for further development of somatic embryos into plants, as most of the SE-derived _AHL15_ overexpressing plants were still diploid.

Disruption of heterochromatin in human mitotic cells leads to mis-segregation of chromosomes[46–49] and cellular polyploidization[50]. We hypothesized that heterochromatin disruption and more global chromatin decondensation in dividing _p35S:AHL15_ cotyledon cells might contribute to endomitosis resulting in polyploid somatic embryo progenitor cells. Compared to chromosome segregation in wild-type dividing cotyledon cells (Fig. 5h), chromosome segregation in dividing _p35S:AHL15_ cotyledon cells lagged behind (Fig. 5i) and binucleate cells (Fig. 5j, k) could be detected in cotyledons of _p35S:AHL15_ IZE 7 days

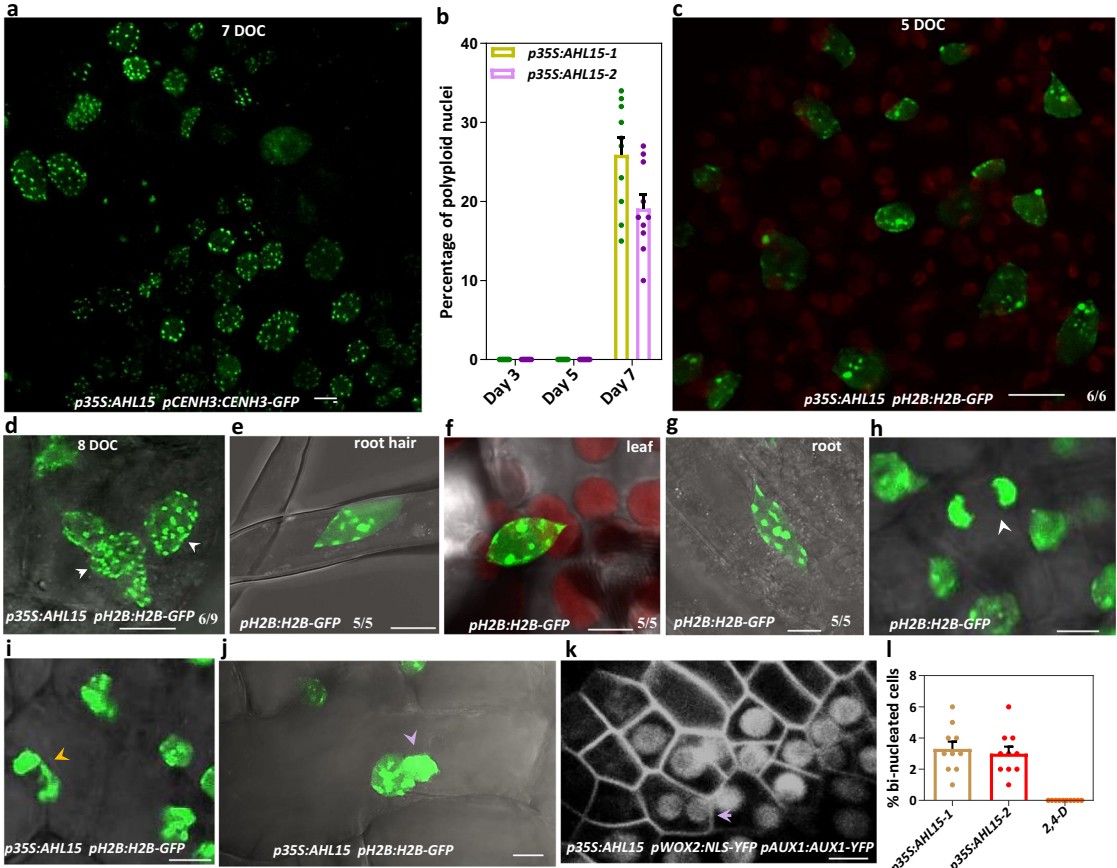

**Fig. 5 _AHL15_ overexpression-induced SE results in polyploidy by endomitosis due to chromosome mis-segregation. a** Confocal image of polyploid cells detected by CENH3-GFP-mediated centromere labeling in an embryonic structure developing on a cotyledon of a _35S:AHL15_ IZE after 7 days of culture (DOC). **b** Percentage of polyploid nuclei (more than 11 detectable CENH3-GFP-labeled centromeres) in cotyledon cells of _p35S:AHL15_ IZEs after 5 or 7 DOC. Dots indicate the values of ten biological replicates per plant line, with about 200 nuclei analyzed per replicate, bar indicates the mean, and error bars the s.e.m. **c, d** H2B-GFP-labeled chromocenters in nuclei of cotyledon cells of _p35S:AHL15_ IZEs after 5 (**c**) or 8 (**d**) DOC. White arrowheads (**d**) indicate cells with a duplicated number of chromocenters. **e–g** Confocal images of H2B-GFP labeled chromocenters in endoreduplicated nuclei of wild-type root hair (**e**), leaf (**f**), or root epidermis (**g**) cells. **h–j** Confocal microscopy analysis of chromosome segregation in dividing protodermal cells in cotyledons of _p35S:AHL15_ IZEs after 8 DOC using the _pH2B:H2B-GFP_ reporter. The white arrowhead indicates normal chromosome segregation during anaphase (**h**), the yellow arrowhead indicates mis-segregation of chromosomes during anaphase (**i**), and the magenta arrowhead indicates a bi-nucleated cell (**j**). **k** Confocal microscopy image of a cotyledon of a _p35S:AHL15 pWOX2:NLS-YFP pAUX1:AUX1-YFP_ IZE. _pWOX2:NLS-YFP_ and _pAUX1:AUX1-YFP_ reporters were used to mark embryonic nuclei and plasma membranes, respectively. The magenta arrowhead indicates a bi-nucleated cell in an area of cells with WOX2-YFP-marked embryo cell fate. **l** Percentage of bi-nucleated cells, as detected by H2B-GFP labeling, in protodermal cotyledon cells of wild-type or _p35S:AHL15_ IZEs after 7 DOC on medium with or without 2,4-D, respectively. Dots indicate the values of ten biological replicates per plant line (two _35S:AHL15_ lines were included) with about 200 nuclei analyzed per replicate, bar indicates the mean, and error bars the s.e.m. **a–k** Similar results were obtained from two independent experiments. Size bar indicates 6 μm. **d–k** Images show a merge of the transmitted light and the GFP channel (**d–j**), or the GFP (**a, c**) or YFP (**k**) channel alone. **c–g** Numbers in images indicate the frequency with which similar images from different samples were obtained.

after culture. The observation that binucleate cotyledon cells expressed the _pWOX2:NLS-YFP_ embryo marker (Fig. 5k, i) confirmed that such cells can adopt embryo identity and develop into polyploid somatic embryos. By contrast, we did not observe any binucleate cells in _pWOX2:NLS-YFP_ expressing cotyledon tissues of 2,4-D treated IZEs (Fig. 5l and Supplementary Fig. 17). Taken together, we conclude that heterochromatin disruption in _p35S:AHL15_-induced embryonic cotyledon cells leads to chromosome mis-segregation, the formation of binucleate cells and finally to cellular polyploidization coinciding with the development of polyploid somatic embryos.

**TSA treatment or long heat stress exposure induce polyploidy during SE initiation.** Inhibition of histone deacetylases by trichostatin A (TSA), a chemical that induces chromatin

decondensation[51], has been reported to promote the transition of plant somatic cells and microspores toward an embryonic state[52,53]. We therefore asked whether the decondensation of chromatin by TSA treatment induces polyploidy during SE. To test this, _pCENH3:CENH3-GFP_ IZEs cultured for 7 days on medium containing 2,4-D were transferred to new medium containing 2,4-D and TSA. We observed a variable number of CENH3-GFP-labeled centromeres (6–8, 12–15, and 25–30) per cell in embryonic cells 2 days after TSA treatment, indicating that polyploidization occurred (Fig. 6a). By contrast, we did not observe such a variable number of centromeres in embryonic cells on medium containing only 2,4-D (Fig. 6a). Quantification of the number of nuclei with more than 11 detectable centromeres confirmed that TSA treatment resulted in polyploidisation in 2,4-D-induced embryonic cells (Fig. 6b). Also long heat stress (LHS) exposure has been shown to induces heterochromatin

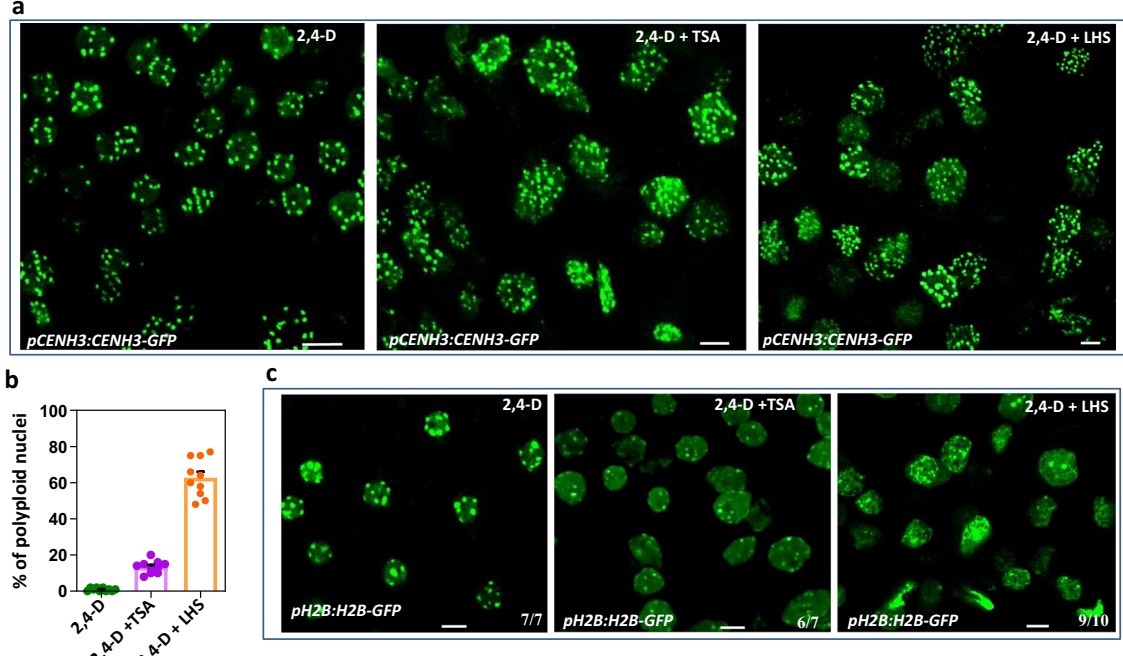

**Fig. 6 TSA treatment or heat stress result in polyploidy in 2,4-D induced embryonic cells by promoting heterochromatin decondensation. a** Confocal image of CENH3-GFP-labeled nuclei in protodermal cells at the adaxial side of IZE cotyledons after 9 DOC. IZEs were cultured for 9 days on medium with 2,4-D (left), or after 7 days on 2,4-D medium transferred for two days to medium with 2,4-D and 1 μM TSA (middle), or exposed for 2 days to long heat stress (LHS) at 38 °C (right). **b** Percentage of polyploid nuclei (based on more than ten detectable CENH3-GFP-labeled centromeres) in cotyledon cells (shown in **a**). Dots indicate the values of ten biological replicates per treatment with about 200 nuclei analyzed per replicate, bar indicates the mean, and error bars the s.e.m. **c** Visualization of heterochromatin decondensation in *pH2B:H2B-GFP* IZEs after 9 DOC on medium containing 2,4-D (left), or after 7 DOC on 2,4-D medium followed by 2 days on 2,4-D and 1 μM TSA (2,4-D + TSA, middle), or by 2 days incubation at 38 °C (24D + LSH, right). Size bars indicate 6 μm. **a–c** Similar results were obtained from two independent experiments. **c** Numbers in images indicate the frequency with which similar images from different samples were obtained.

decondensation in Arabidopsis leaves[54]. Incubation of *pCENH3:CENH3-GFP* IZEs, which were precultured for 7 days on medium containing 2,4-D, for 48 h at 38 °C also resulted in cellular polyploidization in the 2,4-D-induced embryonic cells, even at a higher rate than with TSA treatment (Fig. 6a, b). No polyploidy was observed in root cells treated with TSA or LHS (Supplementary Fig. 19), suggesting that TSA treatment or LHS exposure only results in polyploidisation in 2,4-D induced embryonic cells.

About 10 or 40% of plants regenerated from the embryonic cells treated with respectively TSA or LHS developed large flowers (Supplementary Fig. 18a). Visualization of the number of CENH3-GFP-labeled centromeres in root cells showed that the plants with large flowers were indeed polyploid (Supplementary Fig. 18b).

Comparison of the heterochromatin state in IZE cotyledon cells using the H2B-GFP reporter revealed a significant decrease of H2B-GFP signal in nuclei of cells receiving 2,4-D +TSA treatment or 2,4-D + LHS exposure compared to the 2,4-D control (Fig. 6c). Together our data show that the induction of embryonic identity by either *AHL15* overexpression or 2,4-D + TSA or LHS treatment in Arabidopsis IZE cotyledon cells coincides with strong heterochromatin decondensation, which frequently leads to cellular polyploidization and thus to the production of polyploid somatic embryos.

## Discussion

The herbicide 2,4-D is extensively used for SE induction in a wide range of plant species, including *Arabidopsis*. In *Arabidopsis*, SE can also be induced on IZEs or seedlings in the absence of 2,4-D

treatment by the overexpression of specific transcription factors, such as the AIL transcription factor BBM[13,29]. In this study, we showed that AHL15 adds to the list of nuclear proteins whose overexpression induces somatic embryos on IZEs and seedlings in the absence of 2,4-D. In line with this observation, *AHL15* and several of its homologs are upregulated and required for SE induction upon 2,4-D treatment. Furthermore, they are required for efficient *BBM*-induced SE as downstream targets of BBM. Like BBM and other members of the AIL family[10], AHL15 is expressed in the early zygotic embryo and acts redundantly with AHL family members to maintain embryo development.

AT-hook motif-containing proteins are generally considered to be chromatin architecture factors[17,31–33,55]. Studies in animals have shown that chromatin decondensation precedes the induction of pluripotent stem cells and their subsequent differentiation[56]. In the *Arabidopsis* zygote, predominant decondensation of the heterochromatin configuration is likely to contribute to the totipotency of this cell[57]. Our data indicate that *AHL15* overexpression induces a global reduction of the amount of heterochromatin in induced somatic embryonic cells, whereas *ahl* loss-of-function mutants show enhanced heterochromatin formation in in vitro cultured explants, correlating with a reduced embryonic competence of their explants. Based on our results, we suggest a model in which chromatin opening is required for the acquisition of embryonic competence in somatic plant cells (Fig. 7). In this model, chromatin opening is mediated by upregulation of *AHL* gene expression, which can be achieved by *35S* promotor-driven overexpression.

During the S phase of the cell cycle, eukaryotic cells duplicate their chromosomes after which the mitosis machinery ensures

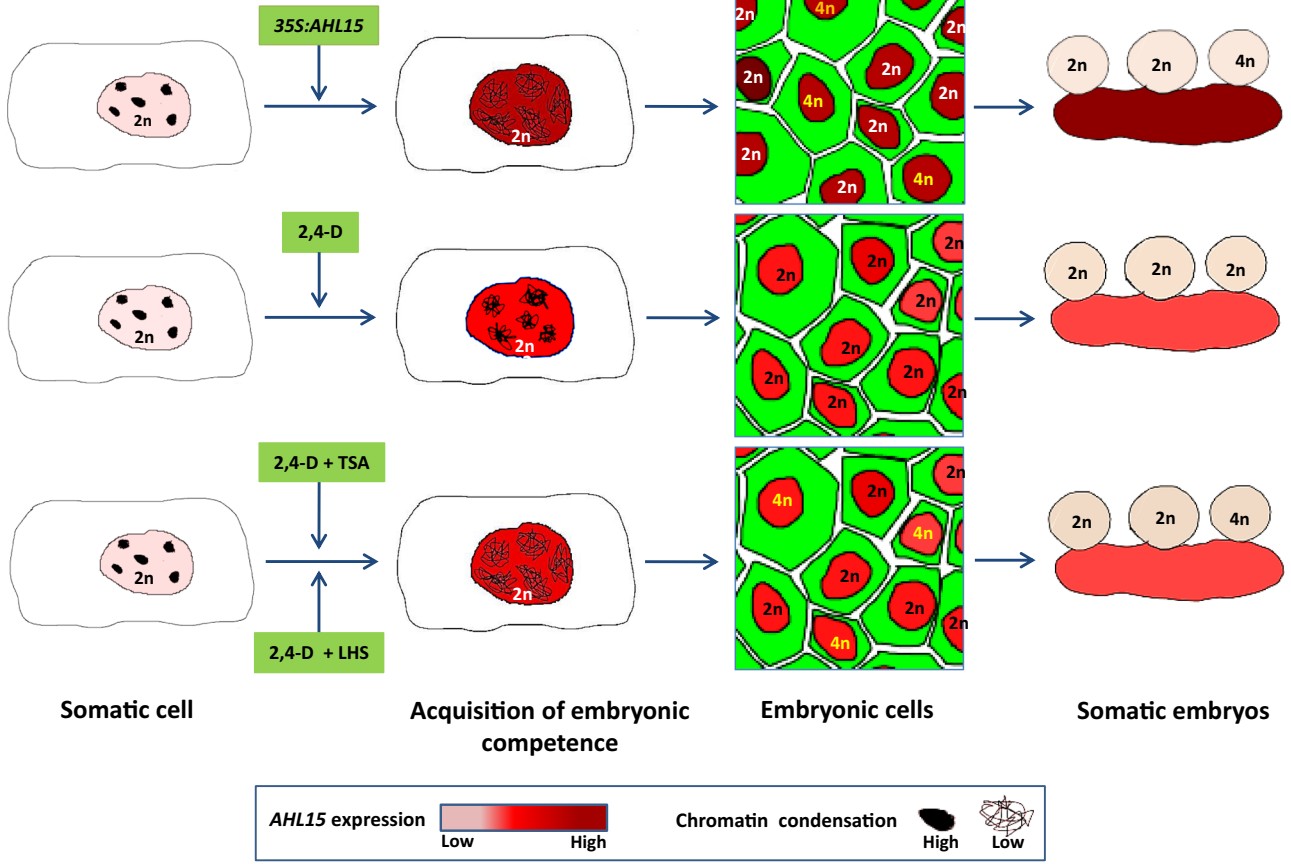

**Fig. 7 Cellular polyploidization occurs during somatic-to-embryonic reprogramming due to strong chromatin decondensation.** *AHL15* overexpression induces strong chromatin decondensation, coinciding with the induction of embryonic competence in somatic cells. This prevents chromosome segregation in some cells, leading to endomitosis events that give rise to polyploid embryonic cells and subsequently to polyploid somatic embryos. By contrast, 2,4-D treatment leads to enhanced *AHL15* expression that is sufficient to induce somatic-to-embryonic reprogramming, but insufficient to induce the level of chromatin decondensation needed for endomitosis. However, when 2,4-D treatment is combined with a treatment that induces strong chromatin decondensation (TSA or heat stress), endomitosis does occur, giving rise to polyploid somatic embryos.

that the sister chromatids segregate equally over the two daughter cells. However, some cell types do not separate the duplicated chromosomes, leading to a polyploid state known as endopolyploidy[58]. In plants, endopolyploidy is commonly classified either as endomitosis or endoreduplication[58]. Endoreduplication occurs during cellular differentiation, where chromosomes are duplicated but do not segregate, leading to the formation of polytene chromosomes[44]. By contrast, during endomitosis sister chromatids are separated, but the last steps of mitosis including nuclear division and cytokinesis are skipped, generally leading to a duplication of the chromosome number. In this work, we show that polyploid cells resulting from endomitosis can be specifically detected during *p35S:AHL15* induced SE.

Previous studies have shown that defects in heterochromatin condensation in animal cells lead to mis-separation of chromosomes during mitosis[46–49], and that mis-segregation of chromosomes subsequently leads to cellular polyploidisation[50]. In our experiments, we found a high reduction of heterochromatin coinciding with mis-segregation of chromosomes in in vitro-cultured *p35S:AHL15* cotyledon cells. Consistent with the strong conservation of chromosome segregation mechanisms between animal and plant cells[59], we propose that cellular polyploidisation in *p35S:AHL15* embryonic cells is caused by an ectopic AHL15-mediated reduction in chromosome condensation during mitosis, which results in chromosome mis-segregation. The observation that polyploid embryos and plants are not obtained after 2,4-D treatment or by *BBM* overexpression suggests that in these somatic

embryos *AHL15* expression levels are finely balanced to prevent strong heterochromatin decondensation and subsequent chromosome mis-segregation (Fig. 7). The fact that 2,4-D culture can lead to polyploid somatic embryos when combined with treatments that enhance chromatin decondensation (TSA or LHS, Fig. 7) provides further evidence that polyploid cells are formed when strong chromatin decondensation coincides with cell proliferation.

A low frequency of polyploid plants derived from somatic embryo cultures has been reported[8,60–63], but the molecular and cytological basis for this genetic instability in relation to in vitro embryogenesis has not been described. Since SE requires cell division, these genome duplications most likely have been caused by endomitosis and not by endoreduplication. Until now, endomitosis in plants has been described as a result of defective cytokinesis due to aberrant spindle-plate or cell-plate formation[42,64,65]. Our results show that chromosome mis-segregation by *AHL15* overexpression-induced chromatin decondensation provides an alternative mechanism of endomitosis in plants, possibly acting not only during SE but also during gametogenesis or zygotic embryogenesis as infrequent environmentally-induced events, that could lead to genome duplication-enabled speciation during evolution. Environmentally-induced or chemically-induced *AHL15* overexpression, combined with TSA treatment or LHS exposure when needed, might also provide an alternative method for chromosome doubling in cultured embryos derived from haploid explants, such as egg cells or microspores[66] and for the production of polyploid crops.

## Methods

**Plant material and growth conditions**. T-DNA insertion mutants *ahl15* (SALK_040729) and *ahl19* (SALK_070123) were obtained from the European Arabidopsis Stock Center (http://arabidopsis.info/). Primers used for genotyping are described in Supplementary Table 1. The reporter lines *pH1.1:H1.1-GFP*[36], *pCENH3:CENH3-GFP*[41] and *pH2B:H2B-GFP*[41], *pWOX2:NLS-YFP*[67], and *pAUX1: AUX1-YFP*[68] have been described previously. For in vitro plant culture, seeds were sterilized in 10% (v/v) sodium hypochlorite for 12 min and then washed four times in sterile water. Sterilized seeds were plated on MA medium[69] containing 1% (w/v) sucrose and 0.7% agar. Seedlings, plants, and explants were grown at 21 °C, 70% relative humidity, and 16 h photoperiod.

**Plasmid construction and plant transformation**. The *35S:AHL15* construct was generated by PCR amplification of the full-length *AHL15* (At3g55560) cDNA from ecotype Columbia (Col-0) using primers 35S:AHL15-F and 35S:AHL15-R (Supplementary Table 1). The resulting PCR product was cloned as a *SmaI/Bgl*II fragment into the *p35S:3'OCS* expression cassette of plasmid pART7, which was subsequently cloned as *Not*I fragment into the binary vector pART27[70]. To generate the other overexpression constructs, the full-length cDNA clones of *AHL19* (At3g04570), *AHL20* (At4g14465), and *AHL29* (At1g76500) from *Arabidopsis* Col-0 were used to amplify the open reading frames (ORFs) using primers indicated in Supplementary Table 1. The ORFs were cloned into plasmid pJET1/blunt (GeneJET™ PCR Cloning Kit, #K1221), and next transferred as *Not*I fragments to binary vector *pGPTV 35S:FLAG*[71]. To generate the *pAHL15:AHL15-GUS* and *pAHL15: AHL15-tagRFP* translational fusions, a 4 kb fragment containing the promoter and the full coding region of *AHL15* was amplified using PCR primers AHL15-GUS-F and AHL15-GUS-R (Supplementary Table 1), and inserted into pDONR207 using a BP reaction (Gateway, Invitrogen). LR reactions were carried out to fuse the 4 kb fragment upstream of GUS and *tagRFP* in respectively destination vectors pMDC163[72] and pGD121[73]. The artificial microRNA (amiR) targeting *AHL20* was generated as described by Schwab and colleagues[74] using oligonucleotides I-IV miR-a/s AHL20 (Supplementary Table 1). The fragment of the *amiRAHL20* precursor was amplified using PCR primers amiRNA AHL20-F and AHL20-R (Supplementary Table 1), and subsequently introduced into the entry vector pDONR207 via a BP reaction (Gateway, Invitrogen). The *amiRAHL20* precursor was recombined into destination vectors pMDC32[72] downstream of the *35S* promoter via an LR reaction (Gateway, Invitrogen). To generate the *pAHL15:AHL15-ΔG* construct, a synthetic *Kpn*I-*Spe*I fragment (BaseClear) containing the *AHL15* coding region lacking the sequence encoding the Gly-Arg-Phe-Glu-Ile-Leu amino acids in the C-terminal region was used to replace the corresponding coding region in the *pAHL15:AHL15* construct. The *p35S:BBM-GR* construct has been described previously[75]. All binary vectors were introduced into *Agrobacterium tumefaciens* strain AGL1 by electroporation[76] and transgenic *Arabidopsis* Col-0 lines were obtained by the floral dip method[77].

**Somatic embryogenesis**. Immature zygotic embryos (IZEs) at the bent cotyledon stage of development (10–12 days after pollination) or germinating dry seeds were used as explants to induce SE using a previously described protocol[8]. In short, seeds and IZEs were cultured on solid B5 medium[78] supplemented with 5 μM 2,4-D, 2% (w/v) sucrose and 0.7% agar (Sigma) for 2 weeks. Control seeds or IZEs were cultured on solid B5 medium without 2,4-D. To allow further embryo development, explants were transferred to medium without 2,4-D. One week after subculture, the capacity to induce SE was scored under a stereomicroscope as the number of somatic embryos produced from 50 explants cultured on a plate. Three plates were scored for each line. The Student's *t*-test was used for statistical analysis of the data.

**Quantitative real-time PCR (RT-qPCR) and ChIP seq analysis**. To determine the expression of *AHL* genes during SE induction, RNA was isolated from 25 IZEs cultured for 7 days on B5 medium with or without 2,4-D in four biological replicates using a Qiagen RNeasy Plant Mini Kit. The RNA samples were treated with Ambion® TURBO DNA-free™ DNase. To determine the expression of *AHL* genes in 2,4-D treated Col-0 IZEs by qRT-PCR, 1 μg of total RNA was used for cDNA synthesis with the iScript™ cDNA Synthesis Kit (BioRad). PCR was performed using the SYBR-Green PCR Master mix (Biorad) and a CHOROMO 4 Peltier Thermal Cycler (MJ RESEARCH). The relative expression level of *AHL* genes was calculated according to the $2^{-\Delta\Delta Ct}$ method[79], using the without 2,4-D value to normalize and the *β-TUBULIN*-6 (At5g12250) gene as a reference. The gene-specific PCR primers used are described in Supplementary Table 1.

The effect of *BBM* overexpression on *AHL* gene expression was examined by inducing five-day-old *Arabidopsis thaliana* Col-0 and *35S:BBM-GR* seedlings (four biological replicates for each line) for 3 h with 10 μM dexamethasone (DEX) plus 10 μM cycloheximide (CHX). RNA was isolated using the Invitek kit, treated with DNase I (Invitrogen) and then used for cDNA synthesis with the Taqman cDNA synthesis kit (Applied Biosystems). qRT-PCR was performed as described above. The relative expression level of *AHL* genes was calculated according to the $2^{-\Delta\Delta Ct}$ method[79], using the wild-type Col-0 value to normalize and the *SAND* (At2g28390) gene[80] as a reference. The gene-specific PCR primers are listed in Supplementary Table 1.

The ChIP-seq data and analysis was downloaded from GEO (GSE52400). Briefly, the experiments were performed using somatic embryos from either 2,4-D-induced *BBM:BBM-YFP* cultures (with *BBM:NLS-GFP* as a control) or a *35S:BBM-GFP* overexpression line (with *35S:BBM* as a control), as described in the ref. [30].

**Ploidy analysis**. The ploidy level of plants derived from *p35S:AHL15*-induced somatic embryos was determined by flow cytometry (Plant Cytometry Services, Schijndel, Netherlands), and confirmed by counting the total number of chloroplasts in stomatal guard cells and by comparing flower size and or the size of the nucleus in root epidermal cells. The number of chloroplasts in stomatal guard cells was counted for plants derived from 2,4-D-induced and BBM-induced somatic embryos.

**Histological staining and microscopy**. Histochemical β-glucuronidase (GUS) staining of *pAHL15:AHL15-GUS* IZEs or ovules was performed as described previously[81] for 4 h at 37 °C, followed by rehydration in a graded ethanol series (75, 50, and 25%) for 10 min each. GUS stained tissues were observed and photographed using a LEICA MZ12 microscopy (Switzerland) equipped with a LEICA DC500 camera.

DNA staining of wild-type and *p35S:AHL15* seedlings was performed using propidium iodide (PI) according to the protocol described by Baroux et al.[82]. The IZEs were fixed in 1% formaldehyde, 10% DMSO, 2 mM EGTA, 0.1% Tween in 1× phosphate-buffered saline (PBS) 20 for 30 min, followed by three washes in PBT (0.1% Tween 20 in 1× PBS), incubation twice in methanol and twice in ethanol for 2 min each, xylene:ethanol (1:1, v/v) for 30 min, and twice in ethanol for 10 min. The sample was subsequently rehydrated in a series of ethanol (90, 70, 50, and 30%, 5 min each). Following rehydration, tissues were washed twice with PBT for 5 min and incubated in enzyme digestion solution (0.5% driselase [Sigma-Aldrich], 0.5% cellulase [Duchefa Biochemie], 0.5% pectolyase [Duchefa Biochemie] in 50 mM PIPES, 5 mM MgCl2, and 5 mM EGTA) at 37 °C for 120 min. This was followed by two washes in PBT for 10 min each and RNase treatment at 37 °C (100 μg/ml RNase A [Roche] in 1× PBS with 1% Tween 20) for 120 min. The samples were then washed 15 min with PBT containing 1% formamide, and were washed twice in PBS for 15 min each and stained with 10 μg/ml PI in PBS for 30 min. Stained samples were placed on a microscope slide mounted with a cover slip.

To stain nuclei, the samples were incubated for 30 min in 4′,6-diamidino-2-phenylindole (DAPI) staining solution (1 μg/ml DAPI in 1× PBS just before observation.

Immunostaining of cultured IZEs was performed as described previously[83]. The IZEs were fixed in 1% formaldehyde, 10% DMSO, 2 mM EGTA, 0.1% Tween in 1× PBS for 30 min, and subsequently embedded in 5% acrylamide on microscope slides. Samples were then processed by incubating them in ethanol and methanol for 5 min each, xylene:ethanol (1:1, v/v) for 30 min, 5 min in methanol, 5 min in ethanol, and 15 min in methanol:PBT (1:1, v/v), complemented with 2.5% formaldehyde. The samples were then washed with PBT and were incubated in enzyme digestion solution (0.5% driselase [Sigma-Aldrich], 0.5% cellulase [Duchefa Biochemie], 0.5% pectolyase [Duchefa Biochemie] in 50 mM PIPES, 5 mM MgCl2, and 5 mM EGTA) at 37 °C for 120 min. This was followed by two washes in PBT, RNase treatment at 37 °C (100 μg/ml RNase A [Roche] in 1× PBS with 1% Tween 20) for 120 min. After that the samples were permeabilized in PBS with 2% Tween-20 at 4 °C for 120 min. For immunostaining, samples were incubated with the primary antibody against H3K9me2 (1:500 dilution, 07-212, Sigma-Aldrich) for 12 h and subsequently with the goat anti-rabbit IgG (H+L) secondary antibody (1:200 dilution, SAB4600234-125UL, Sigma-Aldrich) for 24 h at 4 °C. The samples were mounted in anti-fading liquid mountant supplemented with 10 μg/ml PI.

For scanning electron microscopy (SEM), seedlings were fixed in 0.1 M sodium cacodylate buffer (pH 7.2) containing 2.5% glutaraldehyde and 2% formaldehyde. After fixation, samples were dehydrated by a successive ethanol series (25, 50, 70, 95, and 100%), and subsequently critical-point dried in liquid CO2. Dried specimens were gold-coated and examined using a JEOL SEM-6400 (Japan).

For morphological studies of embryos, fertilized ovules were mounted in a clearing solution (glycerol:water:chloral hydrate = 1:3:8 v/v) and then incubated at 65 °C for 30 min and observed using a LEICA DC500 microscope (Switzerland) equipped with differential interference contrast (DIC) optics.

The number of chloroplasts in leaf guard cells, the size of the DAPI stained nuclear area in root cells and the number of conspicuous heterochromatin regions of the PI stained nuclei of cotyledon cells were recorded using a confocal laser scanning microscope (ZEISS-003-18533), using a 633 laser and 488 nm LP excitation and a 650–700 nm BP emission filters for chlorophyll signals in guard cells, a 405 laser and 350 nm LP excitation and 425–475 nm BP emission filters for DAPI signals in cotyledon cells, and a 633 laser and 488 nm LP excitation and 600–670 nm emission BP filters for PI signals in cotyledon cells. The relative nuclei area in root cells were determined by measuring the DAPI stained nuclei on confocal images using ImageJ software (Rasband).

Cellular and subcellular localization of AHL15-tagRFP and H2B-GFP or CENH3-GFP protein fusions were visualized using the same laser scanning microscope with a 633 laser and 532 nm LP excitation and 580–600 nm BP emission filters for tagRFP signals, or a 534 laser and 488 nm LP excitation and 500–525 nm BP emission filters for GFP or Anti-rabbit IgG (H+L) signals.

**Reporting summary**. Further information on research design is available in the Nature Research Reporting Summary linked to this article.

## Data availability

All processed data are contained in the manuscript or in the Supplementary Information. Materials generated in this study are available upon reasonable request. Source data are provided with this paper.

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

## Acknowledgements

We thank Gerda Lamers for help with microscopy and are grateful to Johan Pinas (†), Nick Surtel, Ward de Winter, and Jan Vink for their help with plant growth and media preparation. O.K. was supported in part by Grant G14_006.02 to R.O. from Generade, and by the Building Blocks of Life grant 737.016.013 to R.O. from the Netherlands Organisation for Scientific Research. M.C. and P.M. were partially supported through grant IS054064 from the Ministry of Economic Affairs (SenterNovem).

## Author contributions

B.vd.Z. and R.O. conceived the project, all authors designed experiments, and analyzed and interpreted results, O.K. performed the majority of the experiments, A.R. contributed to polyploidisation analysis and generated *pAHL15:AHL15-ΔG* transgenic lines, P.M. generated and analysed *Arabidopsis* lines overexpressing the *AHL* genes, A.H. performed the BBM-related experiments, O.K. and R.O. wrote the manuscript, K.B., B.vd.Z. and A.H. assisted in finalizing the manuscript.

## Competing interests
The authors declare no competing interests
