## [Peer Review File · Nature Communications]

Reviewers' comments:

Reviewer #1 (Remarks to the Author):

The manuscript by Omid Karami et al. ("An Arabidopsis AT-hook motif nuclear protein mediates somatic embryogenesis and coinciding genome duplication") describes the role of ATH15 as a novel inducer of somatic embryogenesis (SE). The authors describe AHL15 to be an effector acting downstream of the known SE-inducing TF BBM which can, upon overexpression, induce SE in immature zygotic embryos (IZEs) independently on the use of the synthetic auxin 2,4-D. The authors then describe the effect of AHL15 on heterochromatin de-condensation and reduction of chromocenter size. The authors demonstrate that plants generated from 35S::ATH15-induced SEs are polyploid and suggest that the overexpression of AHL15, unlike overexpression of BBM or the use of 2,4-D, induces polyploidization through endomitosis.

In general, I find the identification of a novel SE-inducing factor interesting and potentially helping to elucidate the molecular pathways acting during 2,4-D- or BBM-induced SE. I however lack more thorough experimental support for some of the conclusions made, especially concerning the second half of the manuscript where the relation of AHL15 to chromatin arrangement and polyploidization is made. This impression is especially strengthened by mentioning data/results that are not shown in the manuscript, lack of quantitative analysis of most of the data shown in Fig 4 and 5 (and interpretations made in the text not being convincingly supported by the images) and also by the lack of proper controls at several places. As such, although I think the general model suggested by the authors may be plausible, I do not think it is sufficiently evidenced by the presented data.

1. From the beginning, it is unclear why AHL15 was selected out of all the 29 AHL genes present in Arabidopsis (phylogeny shown in Fig S1). Resp. it is unclear why in the initial screen AHL15, 19, 20 and 29 were tested. Although Fig. 3 (later!!) shows that some of these genes were previously identified as a BBM targets by the authors, the initial selection is not justified in an understandable way.
2. To my knowledge, AHL15 has not been described before - I would have therefore expected a more thorough characterization of the gene, its expression in the plant, mutant lines etc. (The results part starts immediately from the description of a 35S::AHL15 lines - it is also unclear how many independent lines were used etc.).
3. Line 110: the authors say that the expression of AHL15 peaks at the bent-cotyledon stage - but no further stages in the plant development are shown and therefore the statement may be misleading (Fig 2).
4. The authors show a dominant negative effect of pAHL15::AHL15-GUS in the ahl15 mutant background (but not in WT background). This is based on the apparent lack of phenotype in ahl15 but lack of segregating ahl15/- pAHL15::AHL15-GUS F2 progeny of the pAHL15::AHL15-GUS x ahl15 cross and presence of aborted F2 seeds in the siliques of the respective parental genotype.
 - Nothing is stated of the direction of the cross and the F1 seed phenotype. Can a transmission defect of the ahl15 allele be ruled out (see also next point)?
 - ahl15 ahl19 amiRAHL20 line is said to show WT ZE-development (line 144): (i) this is not demonstrated in any of the figures! and (ii) it is unclear why single ahl15 is not used for comparison.
 - The aberrant F2 embryo/aborted seed number in is not quantified or shown (line 119 - "around 25% of the embryos show patterning defect" - not shown). In fact, no quantification of the seed phenotype is shown.
 - Although the idea of generating the pAHL15::AHL15-deltaG line is nice, I think the nature of the construct is very different from pAHL15::AHL15-GUS, cannot be directly compared and does not explain the dominant negative behaviour of the GUS-tagged construct. Have the authors tried to make an analogous cross with pAHL15::AHL15-tagRFP (which is also used in the study) or with multiple independent transgenic lines?
5. Line 165 and Fig.3: BBM ChIP-seq data are presented but reference to the original work where the data was generated and analysed is not given either in the text or by the figure - only in the methods part. It is not immediately clear that the data has not been generated by this study.
6. The cytogenetic experiments (Figs 4 and 5) require substantial improvements. I do not see support in the images presented in these figures for the statements in the texts:

- e.g. line 188 – 190, Fig 4a (disruption of heterochromatin in 35S::AHL15 compared to Col-0 visualized by PI staining. The images in Fig 4a are not very representative of the quantification in Fig 4c.
- Fig 4b – H2B-GFP signal seems reduced altogether at day 7 in 35S::AHL15 – is this a technical issue or reproducible effect?
- pH2B::H2B-GFP is taken as marker for chromocenter. Although accumulation of H2B would be expected in chromocenter regions in the context of the nuclear space, it is not a conventional chromocenter marker – have the authors tried to confirm using bona fide chromocenter-marking approaches - like immunostaining for H3K9me2 etc...
- In figs 5; S5a,b; S6b, S7 or S8 no data has been quantified or the quantifications are not shown. All the data in these figures should be quantified and statistically evaluated otherwise the statements in the text are not efficiently supported.
- Fig 5l is missing negative controls. How representative is the image?

7. Lines 199-202 – the logic of the argument (referring to Fig S4) is not clear.

8. It is not clear why the nuclear morphology in all experiments is followed on day 3 and 7, but not on day 0 (induction time).

9. Blue light is used in the experiments to connect heterochromatin condensation to SE efficiency. I think this extrapolation may extend too far for the following reasons:

- Bourbousse et al. 2015 have demonstrated the effect of blue light in heterochromatin condensation during seedling de-etiolation, at which starting point heterochromatin is dispersed. This is very different from an experimental setup where IZEs are dissected and incubated in light. The chromocenter morphology and effect of blue light on chromocenters would first need to be established in WT IZEs to determine the effect.
- Blue light signalling during SE is likely to have a pleiotropic effect and more experiments would need to be conducted to connect (limit) the effect on SE efficiency to heterochromatin condensation.

10. Flow cytometry-based ploidy measurement is said to have been conducted (methods, legend to Table 1) but results are not shown – these should be included.

11. Model: line 316 – 320, Fig. 6: AHL15 and homologs are found to be required for 2,4-D-induced SE (see conclusion line 159 – 161) and induction of AHL genes are suggested to be a key component of BBM-triggered SE (line 176 – 178). At the same time, 2,4-D-induced SE is NOT associated with heterochromatin decondensation (Fig S5) or polyploidy (Table 1), similar to 35S::BBM SE (Table 1).

The authors explain these differences by extensive AHL15 expression in the 35S::AHL15 line compared to native induction during 2,4-D or BBM-induced SE, which is plausible and I agree with the first part of the model in Fig 6. I think however that evidence for chromatin decondensation in 2,4-D or BBM-induced SE to support the second part of the model in Fig 6. is missing.

Reviewer #2 (Remarks to the Author):

The authors found that AHL15 is involved in SE formation through the regulation of heterochromatin. This finding is quite interesting but this present manuscript needs the following revision. For example, the authors should add the control of ChIP and several quantitative data in imaging data. The revision will strength their hypothesis.

1. Figure 1a

The authors claim that AHL-OX seedlings initially small and pale. However, the reader cannot understand their claim from the panels at 2 weeks in Figure 1a. It will be better to show the image of plates or more initial seedlings.

2. Figure 2h and i

This panels are too dark. Add the embryo outline with a white dotted line to the panels.

3. Figure 3a

I believe that this data is statistically meaningful. It is a well-known fact that SE induction rate per IZE individually varies within a certain range. Thus, the comparison among average values is not convincing. To emphasis the fidelity, the authors should show the data with box-and-whisker plots

using raw data of 50 explants.

4. Figure 3c, d

The authors claim that AHL15 expression was specifically enhanced in the cotyledon regions. However, GUS attaining signals seem to be detected in the entire region of seedlings.

5. Figure 3e

The authors claim that the number of abnormal somatic embryos was increased in the triple mutant. However, it is impossible to understand the abnormal SE from this image. Add the enlarged image to show a representative abnormal SE and a quantitative data to show the significant increase.

6. q-PCR and ChIP-seq data

Unfortunately, the present q-PCR and ChIP-seq data lack the control. BBM can bind these upstream regions of three AHL genes. Add q-PCR and ChIP-seq data with the upstream region of other AHL genes, which BBM does not bind. The comparison among AHL genes with and without BBM binding will reinforce their expression regulation by BBM.

7. Cotyledon cells

In the later parts, the authors show the cytological data of nuclei in cotyledon cells. However, the cotyledon mainly consists of two different cells, pavement cells of epidermis and mesophyll cells. It is known that these two type of cells exhibit the different chromatin condensation and endoreduplication. Clearly describe the cell materials for these analyses.

8. Figure S3

There is no description on Figure S3. Refer Figure S3 in the text. This merged panel has a problem because tag-RFP signals enhanced in the merged panel compared to single tag-RFP panel. The color intensity should be the same.

9. Figure S4

The signals of H2B seem to be overemphasized because the nucleoplasmic localization of H2B is lost specially in nuclei of the lower region of this panel.

10. Figure 4

It is impossible to understand the remarkable disruption of heterochromatin from the present Figure 4a. Add the enlarged image to exhibit the disruption of heterochromatin clearly. Add the quantitative data to show the diffusion of heterochromatin in Figure 4b. The authors will easily evaluate the diffusion with Image J or similar software. For each replicates 10 nuclei in the quantification of Figure 4c is too small. The quantification with at least more than 50 nuclei is convincing. Explain the difference between days after culture and DAP in the figure legend.

12. Figure S6b

Add the graph to show heterochromatin decondensation quantitatively.

11. Figure S7

There is no data about developed large rosettes with dark green leaves in Figure S7a. Add the graph or dot plot of the chloroplast number, the cell volume and the centromere number to Figure S7.

12. Figure 5

The authors claim that they observed an increase in H2B-GFP-marked chromocenters in 35S:AHL15 cotyledon cells that coincided with polyploidisation events (Figure 5c and d), but there is no data to show the time-course increase in chromocenters. Add the time-course data with a graph. The authors also mention mitotic defects. Add the frequency of lagged behind and binucleate cells as compared to wild type to the text.

13. Discussion

There is no reference of endomitosis. Add the references including Iwata et al. (2011) Plant Cell.

14. Methods

Ref 37 did not describe pH2B:H2B-GFP but pH2B:H2B-CFP.

15. Table 1

Add the discussion about the difference or variety among lines in Table 1.

Reviewer #3 (Remarks to the Author):

The ectopic expression of a single regulator of embryogenesis such as BABY BOOM (BBM) or LEAFY COTYLEDON (LEA) has been shown to trigger the initiation of embryogenesis in somatic tissue and cells. Here, the gene *ahl15* is shown to induce somatic embryogenesis in Arabidopsis cotyledons upon indiscriminate overexpression using the strong promoter 35S. AHL15 is expressed in zygotic embryos but seems non-essential as normal embryos are formed in *ahl15* mutants and an amiRNA_{ahl} knock down line. However, the *ahl15* mutant does not tolerate the introduction of an expression cassette producing AHL15-GUS. This dominant negative effect on embryogenesis occurs only in the *ahl15* mutant background. An AHL15-GUS construct that lacks a domain (PPC) involved in AHL protein interaction with other transcription factors also exhibits a dominant negative impact. Hence, the PPC domain is not involved and the mechanism by which the AHL15-GUS construct leads to a dominant negative phenotype. The issue remains unresolved in the paper. This is a pity because it would give insight into the function of the AHL proteins in controlling embryogenesis. A typical effect of GUS fusions is that the chimeric protein is no longer moving to the nucleus because it surpasses the nuclear pore size exclusion limit. This hypothesis could be tested by localizing a GFP-fusion product.

AHL15 and other members of the AHL family are involved in somatic embryogenesis induced by 2,4D auxin stimulation: mutants show reduced efficiency and the genes are upregulated during the SE program. The AHL15-GUS construct obstructs the SE process in the *ahl15* mutant background resulting in a strong reduction in SEs formed. Line 156 describes that the SE test was done using the *ahl15* mutant in the pAHL15:AHL15-GUS background. I presume it is meant the *ahl15*/+ pAHL15:AHL15-GUS, as no *ahl15* pAHL15:AHL15-GUS homozygous lines could be isolated (line 116) and because that is how it is shown in figure 3a. This result goes against the earlier described results because the *ahl15*/+ pAHL15:AHL15-GUS line produces functional AHL15 and under those conditions AHL15-GUS should not cause a dominant negative phenotype (co-expression of pAHL15:AHL15 in *ahl15* pAHL15:AHL15-GUS does not show dominant negative effects; line 125). Perhaps I misunderstood figure 3a and the labeling should be unambiguously explaining the genotype analysed.

The promoters of AHL15, 19 and 20 bind to BBM, a well known transcription factor that can drive autonomous SE induction when overexpressed. AHL15 and 20 expression is enhanced by 35S:BBM and these genes are required for BBM overexpression induced SE.

AHL15 appears to stimulate chromatin decondensation: overexpression leads to less heterochromatin staining. An independent link between chromatin condensation level and SE is shown through the application of blue light for which it is known that it induced heterochromatin condensation in cotyledons.

The overexpression of AHL15 causes a second phenotype, namely it leads to increased ploidy. The increase is linked with the SE process and does not show e.g. in roots where SEs were not reported to occur (it would be better to explicitly indicate (show data) whether root tissue is devoid of SE induction. The increased ploidy is the result of endomitosis as the number of chromocenters approximately doubles, which is distinct from endoreduplication driven increased ploidy. The mechanism by which overexpression of AHL15 stimulates endomitosis is suggested to be based on chromatin decondensation. Disruption of chromatin condensation in human mitotic cells leads to cellular polyploidisation. A similar phenomenon has not been shown in plants so far and would be simply tested by applying drugs that are affecting chromatin condensation to a line carrying a chromocenter reporter.

The impact of overexpression of AHL15 on chromatin decondensation and ploidy increase sets it

apart from BBM and LEC transcription factors for which no chromatin effect was reported yet also induce SE. That BBM is not inducing a ploidy increase might be because AHL15 expression is not as strong as when driven by 35S. The here shown link between overexpression of AHL15 and an increase in ploidy during SE induction is a surprising phenomenon of interest. Tissue culture is often associated with an increase in ploidy, especially when tissue transforms into callus showing irregular cell division. Based on the findings, one would expect that drugs causing heterochromatin decondensation would induce a similar increase in ploidy in dividing cells. However, it is equally possible that the cytokinesis defect induced by AHL15 is contributing to ectopic cell divisions, as was reported for a callose synthase *gsl8* mutant producing cone shaped protuberances on cotyledons (Saatian et al., 2018; BMC Plant Biol; 18: 295.). More evidence should be provided that heterochromatin decondensation is leading to endomitosis events that explains the increase in ploidy during SE.

Responses to Reviewers' Comments

We thank the reviewers for their substantial effort in reviewing the manuscript. The comprehensive and critical comments and suggestions provide a vital help for improving the manuscript. Below is our point-to-point response to the questions.

Comments from reviewer #1:

General comments

The manuscript by Omid Karami et al. (“An Arabidopsis AT-hook motif nuclear protein mediates somatic embryogenesis and coinciding genome duplication”) describes the role of ATH15 as a novel inducer of somatic embryogenesis (SE). The authors describe AHL15 to be an effector acting downstream of the known SE-inducing TF BBM which can, upon overexpression, induce SE in immature zygotic embryos (IZEs) independently on the use of the synthetic auxin 2,4-D. The authors then describe the effect of AHL15 on heterochromatin de-condensation and reduction of chromocenter size. The authors demonstrate that plants generated from 35S::ATH15-induced SEs are polyploid and suggest that the overexpression of AHL15, unlike overexpression of BBM or the use of 2,4-D, induces polyploidization through endomitosis.

In general, I find the identification of a novel SE-inducing factor interesting and potentially helping to elucidate the molecular pathways acting during 2,4-D- or BBM-induced SE. I however lack more thorough experimental support for some of the conclusions made, especially concerning the second half of the manuscript where the relation of AHL15 to chromatin arrangement and polyploidization is made. This impression is especially strengthened by mentioning data/results that are not shown in the manuscript, lack of quantitative analysis of most of the data shown in Fig 4 and 5 (and interpretations made in the text not being convincingly supported by the images) and also by the lack of proper controls at several places. As such, although I think the general model suggested by the authors may be plausible, I do not think it is sufficiently evidenced by the presented data.

Comment 1:

From the beginning, it is unclear why AHL15 was selected out of all the 29 AHL genes present in Arabidopsis (phylogeny shown in Fig S1). Resp. it is unclear why in the initial screen AHL15, 19, 20 and 29 were tested. Although Fig. 3 (later!!) shows that some of these genes were previously identified as a BBM targets by the authors, the initial selection is not justified in an understandable way.

Authors' response:

In the revised manuscript, we have described shortly how *AHL15* was discovered, and that also *AHL19* and *AHL20* and *AHL29* were tested because they represent the two closest paralogs and a more distant one.

Comment 2:

To my knowledge, AHL15 has not been described before - I would have therefore expected a more thorough characterization of the gene, its expression in the plant, mutant lines etc. (The results part starts immediately from the description of a 35S::AHL15 lines – it is also unclear how many independent lines were used etc.).

Authors' response:

At the moment of initial submission of this manuscript, another paper from our group describing details about AHL15, such as the expression pattern during plant development and mutant lines, was under revision. In the meantime, this paper has been accepted for publication in Nature Plants (see reference below), and in our revised manuscript we refer to this publication:

1- Karami, O., Rahimi, A., Khan, M., Bemer, M., Hazarika, R.R., Mak, P., Compier, M., van Noort., V. & Offringa, R. A (2020) A suppressor of axillary meristem maturation promotes longevity in flowering plants. Nature Plants, 6, 368–376

In the manuscript, we describe now that 9 of the 50 *p35S::AHL15* lines tested produced somatic embryos on seedling cotyledons. In table 1 ploidy levels have been tested in SE-derived plants from several independent *p35S::AHL15* lines. In most other experiments several independent *p35S::AHL15* lines have been tested and data were collected and are presented from two independent *p35S::AHL15* lines.

Comment 3:

Line 110: the authors say that the expression of AHL15 peaks at the bent-cotyledon stage – but no further stages in the plant development are shown and therefore the statement may be misleading (Fig 2).

Authors' response:

In our recent paper in Nature Plants (Karami et al., 2020) we show *AHL15* expression analysis at later developmental stages. However, we agree with the reviewer that in the context of the current manuscript this may be a misleading statement, and have removed it. Now it states “Expression analysisshowed that *AHL15* is expressed in zygotic embryos (ZEs) from the 4 cell embryo stage onward”.

Comment 4:

The authors show a dominant negative effect of *pAHL15::AHL15-GUS* in the *ahl15* mutant background (but not in WT background). This is based on the apparent lack of phenotype in *ahl15* but lack of segregating *ahl15/-pAHL15::AHL15-GUS* F2 progeny of the *pAHL15::AHL15-GUS* x *ahl15* cross and presence of aborted F2 seeds in the siliques of the respective parental genotype.

1- Nothing is stated of the direction of the cross and the F1 seed phenotype. Can a transmission defect of the *ahl15* allele be ruled out (see also next point)?

Authors' response:

Thank you for pointing out this omission. As indicated in the text now, we have performed reciprocal crosses between the *pAHL15::AHL15-GUS* line and the *ahl15* mutant. Furthermore we state: “Irrespective of the direction in which the cross was made, F1 siliques showed a wild-type phenotype, whereas siliques of *ahl15/+ pAHL15::AHL15-GUS* F2 plants contained brown, shrunken seeds”.

2- *ahl15 ahl19 amiRAHL20* line is said to show WT ZE-development (line 144): (i) this is not demonstrated in any of the figures! and (ii) it is unclear why single *ahl15* is not used for comparison.

Authors' response:

ZE-development in the single *ahl15* and the triple *ahl15 ahl19 amiRAHL20* mutant was analysed and is wild type, as presented in Supplementary Fig 4.

3- The aberrant F2 embryo/aborted seed number in is not quantified or shown (line 119 – “around 25% of the embryos show patterning defect” – not shown). In fact, no quantification of the seed phenotype is shown.

Authors' response:

In the revised manuscript, we now show the quantification of aberrant (brown and shrunken) F2 seeds in *pAHL15::AHL15-GUS* and *ahl15/+ pAHL15::AHL15-GUS* plants in Supplementary Fig 5. The text has been changed to “siliques of *ahl15/+ pAHL15::AHL15-GUS* F2 plants contained around 25% brown, shrunken seeds (Figure 2k and Figure S5) that were unable to germinate. Embryos in these shrunken seeds showed patterning defects and did not develop past the globular stage (Figure 2o)”.

4- Although the idea of generating the *pAHL15::AHL15-deltaG* line is nice, I think the nature of the construct is very different from *pAHL15::AHL15-GUS*, cannot be directly compared and does not explain the dominant negative behaviour of the GUS-tagged construct.

Authors' response:

As indicated in our manuscript, the expression of an AHL protein without the conserved six-amino-acid region in the PPC domain has previously been shown to lead to a dominant negative effect in both Arabidopsis and animal systems (references 19 and 28). In line with this, *ahl15/+ pAHL15::AHL15-ΔG* plants produced siliques with brown shrunken seeds just like *ahl15/+ pAHL15::AHL15-GUS* plants (Figure 2r) and

ahl15/+ pAHL15: AHL15-ΔG seedlings phenocopied ahl15/+ pAHL15: AHL15-GUS seedlings, as is shown in another manuscript that is currently under revision:

Rahimi, A., Karami, O., & Offringa, R. miR156-independent repression of ageing in Arabidopsis by AT-kook motif nuclear proteins. PNAS bioRxiv. doi.org/10.1101/2020.06.18.160234

5- Have the authors tried to make an analogous cross with pAHL15::AHL15-tagRFP (which is also used in the study) or with multiple independent transgenic lines?

Authors' response:

Yes, we did, and we have included these results with the following sentence: "This seemed specific for the AHL15-GUS fusion, as fertile homozygous *ahl15 pAHL15:AHL15-tagRFP* plants showing wild-type development could be obtained for three independent *pAHL15:AHL15-tagRFP* lines."

Comment 5:

Line 165 and Fig.3: BBM ChIP-seq data are presented but reference to the original work where the data was generated and analysed is not given either in the text or by the figure – only in the methods part. It is not immediately clear that the data has not been generated by this study.

Authors' response:

The BBM ChIP-seq data was originally published in A. Horstman, H. Fukuoka, J. Muino, L. Nitsch, C. Guo, P. Passarinho, G. Sanchez-Perez, R. Immink, G. C. Angenent, K. Boutilier. (2015). AINTEGUMENTA-LIKE and HOMEODOMAIN GLABROUS proteins antagonistically control cell proliferation. Development 142: 1-11.

In the revised manuscript, we refer to this reference in the results.

Comment 6:

The cytogenetic experiments (Figs 4 and 5) require substantial improvements. I do not see support in the images presented in these figures for the statements in the texts:

1- e.g. line 188 – 190, Fig 4a (disruption of heterochromatin in 35S::AHL15 compared to Col-0 visualized by PI staining. The images in Fig 4a are not very representative of the quantification in Fig 4c.

Authors' response:

We appreciate the reviewer's comments on these points. We repeated visualization of heterochromatin in *p35S::AHL15* compared to Col-0 cotyledon protodermis cells by PI staining as presented in Fig 4a and Supplementary Fig. 7. In the revised manuscript, the heterochromatin has been quantified based on the number of condensed versus dispersed nuclei (Fig 4b), according to the classification shown in Supplementary Fig 8.

2- Fig 4b – H2B-GFP signal seems reduced altogether at day 7 in 35S::AHL15 – is this a technical issue or reproducible effect?

Authors' response:

We repeated visualization heterochromatin in *p35S::AHL15* compared Col-0 cotyledon protodermis cells by H2B-GFP as presented in Fig 4d. We detected similar dispersed H2B-GFP signals at day 7 in *p35S::AHL15* protodermis cells in 8 of the 10 pictures taken from several independent experiments. To show that images are representative, we have added the frequency of observation where needed.

3- pH2B::H2B-GFP is taken as marker for chromocenter. Although accumulation of H2B would be expected in chromocenter regions in the context of the nuclear space, it is not a conventional chromocenter marker – have the authors tried to confirm using bona fide chromocenter-marking approaches - like immunostaining for H3K9me2 etc...

Authors' response:

As suggested by the reviewer, we have performed immunostaining for H3K9me2 (Fig 4e and Supplementary Fig. 9). In addition, we have confirmed decondensation of heterochromatin in

p35S::AHL15 compared to Col-0 cotyledon protodermis cells by using the *pH1.1:H1.1-GFP* reporter (Fig 4c). To show that images are representative, we have added the frequency of observation where needed.

4-In figs 5; S5a,b; S6b, S7 or S8 no data has been quantified or the quantifications are not shown. All the data in these figures should be quantified and statistically evaluated otherwise the statements in the text are not efficiently supported.

Authors' response:

In the revised manuscript, we now show quantification and statistical analysis (graphs) or the frequency of observation per image for Fig 5, and Supplementary Fig. 5a,b (now Fig. S11), Supplementary Fig. 6b (now Fig. S13), and Supplementary Fig. 7 (now Fig. S14).

Comment 7:

Fig 5l is missing negative controls. How representative is the image?

Authors' response:

The negative control of Fig. 5l (now Fig. 5k) is shown in Supplementary Fig 17. Moreover, we have quantified the % of bi-nucleated cells in *p35S::AHL15 pWOX2:NLS-YFP pAUX1:AUX1-YFP* and 2,4-D treated *pWOX2:NLS-YFP pAUX1:AUX1-YFP* cotyledon protodermis cells in Fig. 5l.

Comment 8:

- Lines 199-202 – the logic of the argument (referring to Fig S4) is not clear.

Authors' response:

To clarify our point, we have adapted the last part of the sentence: “....., suggesting that AHL15 action is not limited to heterochromatin, but that the protein regulates global chromatin decondensation.”

Comment 9:

It is not clear why the nuclear morphology in all experiments is followed on day 3 and 7, but not on day 0 (induction time).

Authors' response:

Days 3 and 7 are provided, because time lapse imaging has shown that the first *AHL15* overexpression-induced cell divisions can be observed at 6 days after culture, and that *pWOX2:NLS-YFP* marked somatic pro-embryos can be observed around 6 to 7 days after culture. So at day 3 there is no somatic embryogenesis yet, whereas at day 7 there is. We have clarified this point by the following text:

“First, by tracking SE induction on *p35S::AHL15* IZEs, we observed that protodermal cells at adaxial regions of cotyledons started to divide around six days after culture (Figure S6A), leading to the formation of *pWOX2:NLS-GFP* expressing prosomatic embryos (Figure S6B). Propidium iodide (PI) staining of chromosomal DNA in cotyledon protodermal cells of *35S::AHL15* IZEs showed a remarkable dispersion of heterochromatin coinciding with the appearance of *pWOX2:NLS-GFP* expressing pro somatic embryos at seven days after culture (Figure 4a and Figure S7)”

In the revised manuscript, we now show the nuclear morphology of *p35S::AHL15* or Col-0 IZE cotyledon protodermis cells on day 2 to 7 after culture by PI staining as presented in Figure 4 and Supplementary Fig. 7.

Comment 10:

Blue light is used in the experiments to connect heterochromatin condensation to SE efficiency. I think this extrapolation may extend too far for the following reasons:

- Bourbousse et al. 2015 have demonstrated the effect of blue light in heterochromatin condensation during seedling de-etiolation, at which starting point heterochromatin is dispersed. This is very different from an experimental setup where IZEs are dissected and incubated in light. The chromocenter

morphology and effect of blue light on chromocenters would first need to be established in WT IZEs to determine the effect.

- Blue light signalling during SE is likely to have a pleiotropic effect and more experiments would need to be conducted to connect (limit) the effect on SE efficiency to heterochromatin condensation.

Authors' response:

We agree with the reviewer that the blue light experiment extends too far, and we have removed it from the revised manuscript.

Comment 11:

Flow cytometry-based ploidy measurement is said to have been conducted (methods, legend to Table 1) but results are not shown – these should be included.

Authors' response:

In the revised manuscript, we have now included examples of flow cytometry results of 2n, 4n and 8n plants derived from AHL15-induced somatic embryos (Supplementary Fig. 15).

Comment 12:

Model: line 316 – 320, Fig. 6: AHL15 and homologs are found to be required for 2,4-D- induced SE (see conclusion line 159 – 161) and induction of AHL genes are suggested to be a key component of BBM-triggered SE (line 176 – 178). At the same time, 2,4-D-induced SE is NOT associated with heterochromatin decondensation (Fig S5) or polyploidy (Table 1), similar to 35S::BBM SE (Table 1).

The authors explain these differences by extensive AHL15 expression in the 35S::AHL15 line compared to native induction during 2,4-D or BBM-induced SE, which is plausible and I agree with the first part of the model in Fig 6. I think however that evidence for chromatin decondensation in 2,4-D or BBM-induced SE to support the second part of the model in Fig 6. is missing.

Authors' response:

We have added results showing that 2,4-D leads to moderate heterochromatin decondensation, based on the reduced chromocenter area in nuclei (Supplementary Fig.11), and that this is in line with a lower level of *AHL15* expression in IZEs on 2,4-D medium compared to *p35S::AHL15* IZEs on medium without 2,4-D. For both situations we cannot distinguish whether the heterochromatin decondensation is required for or merely coincides with the induction of SE. However, based on the strong correlation between the *AHL15* expression levels, the levels of chromatin decondensation and the occurrence of genome duplication, we have decided to keep the second part of the model in Fig 6 (Now is Fig. 7) based on the new data with 2,4-D.

Comments from reviewer #2:

General comments

The authors found that AHL15 is involved in SE formation through the regulation of heterochromatin. This finding is quite interesting but this present manuscript needs the following revision. For example, the authors should add the control of ChIP and several quantitative data in imaging data. The revision will strength their hypothesis.

Comment 1:

Figure 1a

The authors claim that AHL-OX seedlings initially small and pale. However, the reader cannot understand their claim from the panels at 2 weeks in Figure 1a. It will be better to show the image of plates or more initial seedlings.

Authors' Response:

We thank the reviewer for pointing out this omission. In Figure 1a, we now show a comparison of the morphology of 1-, 2-, 3- and 4-week-old wild-type and *p35S:AHL15* seedlings.

Comment 2:

Figure 2h and i

This panels are too dark. Add the embryo outline with a white dotted line to the panels.

Authors' Response:

We have marked the embryo outline in Figures 2h and i as requested with a white dotted line.

Comment 3:

Figure 3a

I believe that this data is statistically meaningful. It is a well-known fact that SE induction rate per IZE individually varies within a certain range. Thus, the comparison among average values is not convincing. To emphasis the fidelity, the authors should show the data with box-and-whisker plots using raw data of 50 explants.

Authors' Response:

In Figure 3a we now show a bar graph indicating the average value, dots indicating the values obtained in 3 independent experiments, and error bars indicating the s.e.m. Moreover, statistically different values are marked by different letter.

Comment 4:

Figure 3c, d

The authors claim that AHL15 expression was specifically enhanced in the cotyledon regions. However, GUS attaining signals seem to be detected in the entire region of seedlings.

Authors' Response:

We repeated the GUS staining experiment, and from the new images provided in figure 3d it is clear that *AHL15* is more strongly expressed following 2,4-D treatment, "specifically in the cotyledon regions where somatic embryos are initiated."

Comment 5:

Figure 3e

The authors claim that the number of abnormal somatic embryos was increased in the triple mutant. However, it is impossible to understand the abnormal SE from this image. Add the enlarged image to show a representative abnormal SE and a quantitative data to show the significant increase.

Authors' Response:

In Figure 3e, we have replaced the image for the triple mutant with a new image clearly showing abnormal somatic embryo development. For all three images in this figure we have indicated the frequency of observation

Comment 6:

q-PCR and ChIP-seq data

Unfortunately, the present q-PCR and ChIP-seq data lack the control. BBM can bind these upstream regions of three AHL genes. Add q-PCR and ChIP-seq data with the upstream region of other AHL genes,

which BBM does not bind. The comparison among AHL genes with and without BBM binding will reinforce their expression regulation by BBM.

Authors' Response:

In the revised manuscript, we have included the ChIP-seq and qPCR data of *AHL29*, as an example of a gene that is not bound or activated by BBM. In the ChIP-seq data, *AHL29* only has a small peak at -1.5 kb, while for the other 3 genes the strong peak of BBM binding is much closer to the transcription start site (at ~ -200-500 bp). Moreover, compared to *AHL29*, the qPCR analysis suggests that *AHL19* expression is induced by BBM, as the induction values are close to being statistically significant.

Comment 7:

Cotyledon cells

In the later parts, the authors show the cytological data of nuclei in cotyledon cells. However, the cotyledon mainly consists of two different cells, pavement cells of epidermis and mesophyll cells. It is known that these two type of cells exhibit the different chromatin condensation and endoreduplication. Clearly describe the cell materials for these analyses.

Authors' Response:

Previous results clearly show that 2,4-D-induced somatic embryos are derived from protodermal/subprotodermal cells at the adaxial side of IZE cotyledons (e.g. Kurczynska et al., 2007, *Planta*). This is also where we observe rapid cell divisions leading to the formation of globular embryos 2-3 days later (Fig. S4). All our observations on global chromatin changes have therefore focussed on protodermal cells at the adaxial side of cotyledons. Were possible, we have clarified this point in the figure legends, the results part and the materials and methods section.

Comment 8:

Figure S3

There is no description on Figure S3. Refer Figure S3 in the text. This merged panel has a problem because tag-RFP signals enhanced in the merged panel compared to single tag-RFP panel. The color intensity should be the same.

Authors' Response:

We thank the reviewer for pointing out this omission. We have added a reference to Supplementary Fig. 3 in the Results section.

Comment 9:

Figure S4

The signals of H2B seem to be overemphasized because the nucleoplasmic localization of H2B is lost specially in nuclei of the lower region of this panel.

Authors' Response:

We have repeated the visualization of heterochromatin using the H2B-GFP reporter in *p35S::AHL15* compared to Col-0 IZEs (now Fig. 4d), and have added the frequency of observation in several independent experiments to the images. Moreover, these observations were confirmed by using the *pH1.1:H1.1-GFP* reporter (Fig 4c) and by immunostaining for the H3K9me2 heterochromatin mark (Fig 4e, Supplementary Fig.)

Comment 10:

Figure 4

It is impossible to understand the remarkable disruption of heterochromatin from the present Figure 4a. Add the enlarged image to exhibit the disruption of heterochromatin clearly. Add the quantitative data to show the diffusion of heterochromatin in Figure 4b. The authors will easily evaluate the diffusion with Image J or similar software. For each replicates 10 nuclei in the quantification of Figure 4c is too small.

The quantification with at least more than 50 nuclei is convincing. Explain the difference between days after culture and DAP in the figure legend.

Authors' Response

We have repeated the visualization of heterochromatin by PI staining in *p35S::AHL15* compared Col-0 IZEs (Fig. 4a and Supplementary Fig. 7 and 8), and, based on a classification in condensed or dispersed nuclei (Supplementary Fig. 8), we have quantified the % of dispersed nuclei in protodermal cells of IZE cotyledons at 2, 3, 4, 5, 6, and 7 days after culture (n = 10, 200 nuclei per replicate). This quantification now clearly shows that heterochromatin decondensation in *p35S::AHL15* cells starts at day 6 and is strong at day 7, thus coinciding with the first cell divisions and the appearance of somatic embryo structures.

Comment 11:

Figure S6b

Add the graph to show heterochromatin decondensation quantitatively

Authors' Response

Based on the comments of reviewer 1, we have omitted these data from the manuscript.

Comment 12:

Figure S7

There is no data about developed large rosettes with dark green leaves in Figure S7a. Add the graph or dot plot of the chloroplast number, the cell volume and the centromere number to Figure S7

Authors' Response

We have added an image of a typical polyploid embryo-derived *p35S::AHL15* plant, showing the large rosette with dark green leaves (Supplementary Fig. S13). In addition, we have added dot plots showing the quantification of the flower width, chloroplast number per guard cell, nucleus size and visible centromere number to Supplementary Fig.7 (now Supplementary Fig. S12)

Comment 13:

Figure 5

The authors claim that they observed an increase in H2B-GFP-marked chromocenters in *35S::AHL15* cotyledon cells that coincided with polyploidisation events (Figure 5c and d), but there is no data to show the time-course increase in chromocenters. Add the time-course data with a graph. The authors also mention mitotic defects. Add the frequency of lagged behind and binucleate cells as compared to wild type to the text.

Authors' Response

We have added a time course showing the percentage of polyploid nuclei in *p35S::AHL15* IZE cotyledon protodermis cells based on the CENH3-GFP marker (Fig. 5b). Together with the graph in Fig. 4b, this shows that chromatin decondensation (strong at day 7) does coincide with polyploidisation (at day 7). In addition, we have added a quantification of the percentage of bi-nucleate cells in the cotyledon protodermis of *p35S::AHL15* IZEs compared to 2,4-D-treated IZEs (Fig. 5l).

Comment 14:

Discussion

There is no reference of endomitosis. Add the references including Iwata et al. (2011) Plant Cell.

Authors' Response

Thank you for pointing out this omission. We have referred to Iwata et al. (2011) Plant Cell in the discussion.

Comment 15:

Ref 37 did not describe pH2B:H2B-GFP but pH2B:H2B-CFP.

Authors' Response

In this manuscript they used the CFP-based construct, but the authors also had a GFP-based construct, and this one was provided to us and used for our experiments.

Comment 16:

15. Table 1

Add the discussion about the difference or variety among lines in Table 1.

Authors' Response

We added the following sentence to explain the difference among lines in Table 1: "This variety among lines most likely relates to the level of *AHL15* overexpression in the different *p35S:AHL15* lines."

Comment 17:

Figure S4

The signals of H2B seem to be overemphasized because the nucleoplasmic localization of H2B is lost specially in nuclei of the lower region of this panel.

Authors' Response

We believe the difference in H2B-GFP signal between nuclei is related to the focal plane. Other than that we are confident that the images support our conclusion that *AHL15* does not specifically co-localize with H2B-marked heterochromatin, but rather is distributed throughout the nucleoplasm.

Comments from reviewer #3:

General comments

The ectopic expression of a single regulator of embryogenesis such as *BABY BOOM* (*BBM*) or *LEAFY COTYLEDON* (*LEA*) has been shown to trigger the initiation of embryogenesis in somatic tissue and cells. Here, the gene *ahl15* is shown to induce somatic embryogenesis in *Arabidopsis* cotyledons upon indiscriminate overexpression using the strong promoter 35S. *AHL15* is expressed in zygotic embryos but seems non-essential as normal embryos are formed in *ahl15* mutants and an *amiRNA^{ahl}* knock down line.

Comment 1:

The *ahl15* mutant does not tolerate the introduction of an expression cassette producing *AHL15-GUS*. This dominant negative effect on embryogenesis occurs only in the *ahl15* mutant background. An *AHL15-GUS* construct that lacks a domain (PPC) involved in *AHL* protein interaction with other transcription factors also exhibits a dominant negative impact. Hence, the PPC domain is not involved and the mechanism by which the *AHL15-GUS* construct leads to a dominant negative phenotype. The issue remains unresolved in the paper. This is a pity because it would give insight into the function of the *AHL* proteins in controlling embryogenesis. A typical effect of *GUS* fusions is that the chimeric protein is no longer moving to the nucleus because it surpasses the nuclear pore size exclusion limit. This hypothesis could be tested by localizing a GFP-fusion product.

Authors' response:

For many nuclear proteins it has been shown that fusions with *GUS* can move to the nucleus (as an example, see Xu et al., 2016; *PLOS Genetics*; DOI:10.1371/journal.pgen.1006263). In addition, if the protein would not move to the nucleus, it would be unlikely to have a dominant negative effect, unless it would keep other *AHL* proteins out of the nucleus. But this is excluded by the fact that we can complement the effect with wild-type *AHL15* (Figure 2m).

Moreover, we crossed 3 independent *pAHL15::AHL15-tagRFP* lines with the *ahl15* loss-of-function mutant and were able to obtain homozygous plants for all three crosses that were fertile and showed wild-type development. This shows that the dominant negative effect in the *ahl15* mutant background is specific for the AHL15-GUS fusion.

Comment 2:

AHL15 and other members of the AHL family are involved in somatic embryogenesis induced by 2,4D auxin stimulation: mutants show reduced efficiency and the genes are upregulated during the SE program. The AHL15-GUS construct obstructs the SE process in the *ahl15* mutant background resulting in a strong reduction in SEs formed. Line 156 describes that the SE test was done using the *ahl15* mutant in the *pAHL15:AHL15-GUS* background. I presume it is meant the *ahl15/+ pAHL15:AHL15-GUS*, as no *ahl15 pAHL15:AHL15-GUS* homozygous lines could be isolated (line 116) and because that is how it is shown in figure 3a. This result goes against the earlier described results because the *ahl15/+ pAHL15:AHL15-GUS* line produces functional AHL15 and under those conditions AHL15-GUS should not cause a dominant negative phenotype (co-expression of *pAHL15:AHL15* in *ahl15 pAHL15:AHL15-GUS* does not show dominant negative effects; line 125). Perhaps I misunderstood figure 3a and the labeling should be unambiguously explaining the genotype analysed.

Authors' response:

We thank the reviewer for pointing out this unclarity. Indeed, IZEs were harvested from *ahl15/+ pAHL15:AHL15-GUS* siliques and incubated on 2,4-D containing medium according to the protocol described in the materials and methods. Following incubation and counting the embryos per explant, we genotyped the explants and only used the score of *ahl15/+ pAHL15:AHL15-GUS* explants. We have clarified this in the text and in the legends of Figure 3.

Comment3:

The promoters of AHL15, 19 and 20 bind to BBM, a well known transcription factor that can drive autonomous SE induction when overexpressed. AHL15 and 20 expression is enhance by 35S:BBM and these genes are required for BBM overexpression induced SE. AHL15 appears to stimulate chromatin decondensation: overexpression leads to less heterochromatin staining. An independent link between chromatin condensation level and SE is shown through the application of blue light for which it is known that it induced heterochromatin condensation in cotyledons.

The overexpression of AHL15 causes a second phenotype, namely it leads to increased ploidy. The increase is linked with the SE process and does not show e.g. in roots where SEs was not reported to occur (it would be better to explicitly indicate (show data) whether root tissue is devoid of SE induction).

Authors' response:

SE was not observed in the other tissues such as root, hypocotyl and leaves. We have mentioned this in text: "No evidence was obtained for polyploidy in root meristems (Figure S16a) or young leaves (Figure S16b) of ZE-derived *p35S:AHL15* plants, nor was polyploidy observed in the 2,4-D-induced non-embryogenic calli found on leaf and root tissues of *p35S:AHL15* plants (Figure S16c, d)."

Comment4:

The increased ploidy is the result of endomitosis as the number of chromocenters approximately doubles, which is distinct from endoreduplication driven increased ploidy. The mechanism by which overexpression of AHL15 stimulates endomitosis is suggested to be based on chromatin decondensation. Disruption of chromatin condensation in human mitotic cells leads to cellular polyploidisation. A similar phenomenon has not been shown in plants so far and would be simply tested by applying drugs that are affecting chromatin condensation to a line carrying a chromocenter reporter.

The impact of overexpression of AHL15 on chromatin decondensation and ploidy increase sets it apart from BBM and LEC transcription factors for which no chromatin effect was reported yet also induce SE. That BBM is not inducing a ploidy increase might be because AHL15 expression is not as strong as when driven by 35S. The here shown link between overexpression of AHL15 and an increase in ploidy during SE induction is a surprising phenomenon of interest. Tissue culture is often associated with an increase in ploidy, especially when tissue transforms into callus showing irregular cell division. Based on the findings, one would expect that drugs causing heterochromatin decondensation would induce a similar increase in ploidy in dividing cells. However, it is equally possible that the cytokinesis defect induced by AHL15 is contributing to ectopic cell divisions, as was reported for a callose synthase *gsl8* mutant producing cone shaped protuberances on cotyledons (Saatian et al., 2018; BMC Plant Biol; 18: 295.). More evidence should be provided that heterochromatin decondensation is leading to endomitosis events that explains the increase in ploidy during SE.

Authors' response:

We appreciate this helpful comment by the reviewer. To show that polyploidisation is linked to heterochromatin decondensation, we have tested trichostatin A (TSA) and long heat stress (LHS) treatment. Both treatments have been shown to induce heterochromatin decondensation in Arabidopsis. In the revised manuscript we now confirm that culturing Arabidopsis IZEs on 2,4-D medium together with TSA treatment or LHS exposure leads to heterochromatin decondensation (Fig. 6 c), and that this coincides with cellular polyploidization in cotyledon protodermis cells (Fig.6 a and b). About 12% (TSA) and 29% (LHS) of the somatic embryos and resulting plants are polyploid (Supplementary Fig. 18). In contrast, as expected from previous results (Table 1, Supplementary Fig. S11), culturing IZEs on 2,4-D medium alone did not lead to polyploidisation, as all somatic embryos and derived plants were diploid.

REVIEWERS' COMMENTS

Reviewer #1 (Remarks to the Author):

In the revised version of the manuscript, the authors have provided substantial amount of support for their hypotheses. Especially the cytogenetic part of the manuscript has been much improved. The current version of the manuscript in my opinion provides a very nice contribution to our understanding of somatic embryogenesis. My previous comments have been well addressed. I have only two minor comments/questions:

1. Line 226 – 230: The fact that AHL15-tagRFP did not co-localize with heterochromatin is considered surprising. This finding would however implement that AHL15 would act only/predominantly on heterochromatin. To my knowledge, it has not been determined whether the heterochromatin decondensation per se is the driver of the cell identity change or it is “merely” a general marker of chromatin decondensation that would also include repressed euchromatin regions (that would not be visible when assessing chromocentres). In such scenario, AHL15 would be expected to generally play role in the nucleus, not only within chromocentres.
2. Fig. S12, line 247: Chromocentres in the defective embryos of the *ahl15* pAHL15::AHL15-GUS (dominant negative line) are said to be much larger. I agree with the statement (based on quantification as proportion of nuclear area in Fig S12c). I noticed however that the nuclei in this line (Fig.S12b) are considerably larger than WT (S12a). With approx. the same number of chromocenters. How representative is this image? Could this indicate endoreduplication and if so, how does it match the general proposed model of AHL15 function?

Reviewer #2 (Remarks to the Author):

The authors performed their additional experiments and added convincing data according to three reviewers' comments. This revised version includes the verification of experimental data by statistical analyses. This version of manuscript will exactly provide new valuable information on the function of AHL15 in SE.

Reviewer #3 (Remarks to the Author):

The authors have provide a rebuttal addressing the comments and criticism and improved the manuscript in accordance.

Response to final comments by reviewer #1

Comment 1: Line 226 – 230: The fact that AHL15-tagRFP did not co-localize with heterochromatin is considered surprising. This finding would however implement that AHL15 would act only/predominantly on heterochromatin. To my knowledge, it has not been determined whether the heterochromatin decondensation per se is the driver of the cell identity change or it is “merely” a general marker of chromatin decondensation that would also include repressed euchromatin regions (that would not be visible when assessing chromocentres). In such scenario, AHL15 would be expected to generally play role in the nucleus, not only within chromocentres.

Authors’ response: This is exactly what we conclude in the next sentence: “suggesting that AHL15 action is not limited to heterochromatin, but that the protein rather regulates global chromatin decondensation”.

Comment 2: Fig. S12, line 247: Chromocentres in the defective embryos of the ahl15 pAHL15::AHL15-GUS (dominant negative line) are said to be much larger. I agree with the statement (based on quantification as proportion of nuclear area in Fig S12c). I noticed however that the nuclei in this line (Fig.S12b) are considerably larger than WT (S12a). With approx. the same number of chromocenters. How representative is this image? Could this indicate endoreduplication and if so, how does it match the general proposed model of AHL15 function?

Authors’ response: In this experiment, we visualized the chromocenters in heart stage zygotic embryos at 6 days after pollination. We did not observe more than 10 detectable CENH3-GFP-labelled centromeres in embryo cells at this stage of development in both ahl15 pAHL15::AHL15-GUS and WT, indicating absence of endoreduplication. We believe that the larger chromocenters in ahl15 pAHL15::AHL15- GUS embryo cells are related to a stronger heterochromatin condensation (which is in line with the function of AHL15 indicated in line 249) leading to defects in embryo development.